# Guided Speculative Inference for Efficient Test-Time Alignment of LLMs

**Jonathan Geuter**
Harvard SEAS
Kempner Institute
jonathan.geuter@gmx.de

**Youssef Mroueh**
IBM Research

**David Alvarez-Melis**
Harvard SEAS
Kempner Institute

## Abstract

We propose *Guided Speculative Inference* (GSI), a novel algorithm for efficient reward-guided decoding in large language models. GSI combines soft best-of-$n$ test-time scaling with a reward model $r(x, y)$ and speculative samples from a small auxiliary model $\pi_S(y \mid x)$. We provably approximate both the optimal tilted policy $\pi_{\beta,B}(y \mid x) \propto \pi_B(y \mid x) \exp(\beta\, r(x,y))$ of soft best-of-$n$ under the base model $\pi_B$, as well as the expected reward under the optimal policy. In experiments on reasoning benchmarks (MATH500, OlympiadBench, Minerva Math, MMLU-STEM, GSM8K) and across different model families, our method achieves higher accuracy than standard soft best-of-$n$ with $\pi_S$ and reward-guided speculative decoding (Liao et al., 2025), and in certain settings even outperforms soft best-of-$n$ with $\pi_B$, while reducing end-to-end latency by up to 28%.

## 1 Introduction

Large language models (LLMs) have demonstrated remarkable performance across diverse generation tasks, with scaling model and data size being the predominant way to reliably enhance their capabilities (Kaplan et al., 2020; Gemini Team, 2024; OpenAI et al., 2024). However, such scaling incurs ever-increasing computational and economic costs, and there is growing evidence that scaling training compute yields diminishing returns (Hernandez et al., 2022; Muennighoff et al., 2023), prompting the need for efficient alternatives.

*Test-time scaling* (Snell et al., 2025; Muennighoff et al., 2025; Zhang et al., 2025) has emerged as a promising direction, which focuses on scaling inference-time rather than training time compute. Various test-time scaling methods, such as best-of-$n$ sampling (Gao et al., 2023; Mroueh & Nitsure, 2025; Beirami et al., 2025) and soft best-of-$n$ sampling (Verdun et al., 2025), have been proposed, all of which achieve improved downstream performance through increasing inference FLOPs. However, users can have constraints on inference compute and latency, and test-time scaling can quickly become prohibitively expensive. This has led to the development of latency-efficient test-time scaling methods such as speculative decoding (Leviathan et al., 2023; Sun et al., 2025), where a small draft model $\pi_S$ accelerates inference from a larger target model $\pi_B$.[1]

Moreover, the goal is oftentimes not only to achieve better downstream performance, but to do so in a way that maximizes the rewards of a given reward function $r(x, y)$ quantifying the quality of a response $y$ given a prompt $x$. Several frameworks for aligning model outputs to a reward model have been proposed, both for training as well as at test-time (Yang & Klein, 2021; Ouyang et al., 2022; Touvron et al., 2023; Mudgal et al., 2024; Huang et al., 2025). Recent work on reward-guided speculative decoding (RSD) (Liao et al., 2025) combines model alignment with speculative decoding from a draft model, though it lacks theoretical guarantees on distributional fidelity.

**Contributions.** In this paper, we introduce a novel test-time algorithm, *Guided Speculative Inference* (GSI), which leverages samples from a draft model $\pi_S$ to (approximately) sample from the

---

[1]We will interchangeably call $\pi_S$ the *draft* or *small* model, and $\pi_B$ the *base* or *target* model.

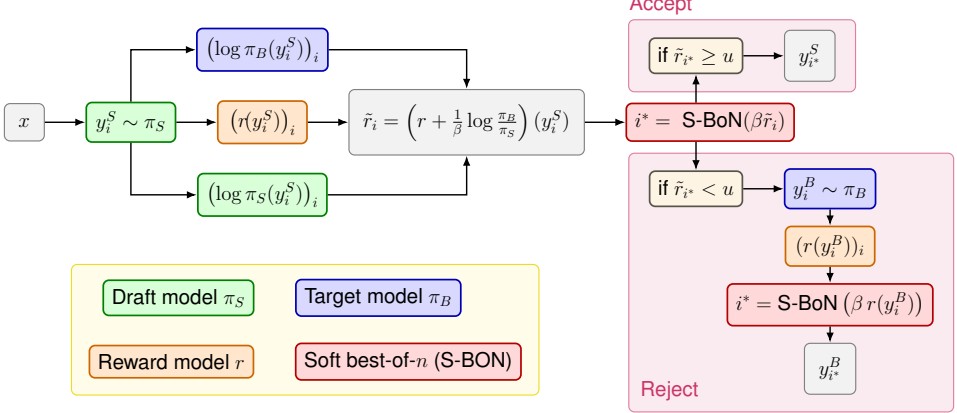

Figure 1: Guided Speculative Inference workflow for one reasoning step. A sample $y_{i*}^S$ generated from the draft model $\pi_S$ is selected with soft best-of-$n$ (S-BoN) with parameter $\beta$ from the *tilted rewards* $\tilde{r}_i$. If its reward lies above a threshold $u$ it is accepted. Otherwise, it is rejected, which triggers resampling from the target model $\pi_B$ with soft best-of-$n$.

base distribution $\pi_B$ aligned to a reward model $r$, namely the tilted distribution (Section 4):

$$\pi_{\beta,B}(y \mid x) \;=\; \frac{\pi_B(y \mid x)\exp(\beta\, r(x,y))}{Z_{\beta,B}(x)}.$$

Importantly, by *tilting* (i.e., adjusting) the rewards $r$ according to the log-likelihoods under both $\pi_B$ and $\pi_S$ (Figure 1), GSI provably approximates this tilted distribution, making it the first speculative test-time scaling method with distributional guarantees to the optimal tilted distribution, to the best of our knowledge. We summarize our contributions as follows:

- We propose a novel test-time scaling algorithm, *Guided Speculative Inference* (GSI), which uses a draft model $\pi_S$ to accelerate inference from a target model $\pi_B$ while aligning responses to a given reward model $r$ (Section 4)

- We prove that GSI enjoys strong theoretical guarantees and provably approximates the optimal tilted distribution (Theorem 1), as well as the expected reward (Theorem 2)

- In extensive experiments on reasoning benchmarks (MATH500, OlympiadBench, Minerva Math, MMLU-STEM, GSM8K) and across model families (Qwen-2.5-Math, Qwen-3) and sizes, we demonstrate that GSI outperforms both reward-guided speculative decoding (Liao et al., 2025) and soft best-of-$n$ sampling with the draft model, and sometimes even soft best-of-$n$ sampling with the target model (Section 5.1)

## 2 RELATED WORK

**Test-Time Scaling.** Inference time compute can be scaled along different axes. Broadly, such methods can be divided into *parallel* and *sequential* approaches. In sequential approaches, the model spends more time on a *single* response and aims to improve it, for example by appending *think tokens* (Muennighoff et al., 2025) or via self-correction (Qu et al., 2024). While sequential approaches can often generate high-quality responses, they don't scale well. Parallel approaches instead scale test-time by parallelizing computations, which typically involves generating multiple responses or reasoning steps at a time. Common parallel approaches include majority voting (Wang et al., 2023) and best-of-$n$ sampling (Mroueh & Nitsure, 2025; Beirami et al., 2025) (see Section 3).

**Speculative Decoding.** Speculative decoding (SD) (Leviathan et al., 2023) accelerates sampling from $\pi_B$ by first drawing proposals from $\pi_S$ and then accepting or rejecting them based on a criterion derived from the ratio $\pi_B/\pi_S$. On rejection, one falls back to direct sampling from $\pi_B$. SD provably samples from the distributions of $\pi_B$. The core idea is that $k$ tokens can be sampled from $\pi_S$ autoregressively, but verified by $\pi_B$ in parallel, thus generating up to $k + 1$ tokens from $\pi_B$ with

a single forward pass of $\pi_B$. Variants of SD include *block verification* (Sun et al., 2025) where sequences of draft tokens are verified jointly instead of token-by-token, and *SpecTr* (Sun et al., 2023) which allows for verification of multiple draft sequences in parallel by framing SD as an optimal transport problem. SD has also been combined with early-exiting (Liu et al., 2024), and Bhendawade et al. (2024) propose using $n$-gram predictions of $\pi_B$ as drafts, which alleviates the need for an auxiliary model.

**Reward-Guided Speculation.** A recent work proposes RSD (reward-guided speculative decoding) (Liao et al., 2025), where samples are generated from $\pi_S$, and a threshold on the reward of the samples from $\pi_S$ determines whether one should accept the sample or resample from $\pi_B$. While this approach shares similarities with GSI, it only provides a guarantee on the expected reward: under the assumption that $\mathbb{E}_{\pi_B}[r(y \mid x)] \geq \mathbb{E}_{\pi_S}[r(y \mid x)]$, RSD satisfies $\mathbb{E}_{\pi_{\mathrm{RSD}}}[r(y \mid x)] \geq \mathbb{E}_{\pi_S}[r(y \mid x)]$, which in the worst case does not yield any improvement over the small model $\pi_S$, and also does not guarantee anything about the policy $\pi_{\mathrm{RSD}}$ itself. As we will see in Section 4, GSI provides guarantees on the induced policy directly. In concurrent work, Cemri et al. (2025) propose SPECS, an algorithm that pairs draft-generated samples with a cascading routine, which determines which model – draft or target – to use in subsequent iterations. Similar to our Theorem 1, they also derive a KL bound with respect to the target distribution. However, their bound requires assuming that the block size (i.e., the length of reasoning steps) tends to infinity, and that the number of samples $n$ and the rejection threshold $u$ are random variables, all of which are approximations that do not hold in practice. Our KL bound in Theorem 1 does not require any such assumptions. Moreover, GSI seems to significantly outperform SPECS on downstream tasks (e.g. up to $11.5\%$ improved accuracy on MATH500). RSD, SPECS, and GSI all have in common that they operate on *reasoning steps* of reasoning models, where each iteration of the algorithm produces a subsequent reasoning step.

## 3 BACKGROUND

Let $\mathcal{V}$ denote a (finite) vocabulary. Let $\mathcal{X} = \bigcup_{n \in \mathbb{N}} \prod_{i=1}^{n} \mathcal{V}$ be the (countable) space of inputs, consisting of finite sequences over the vocabulary (in practice these will be prompts and already generated reasoning steps), and $\mathcal{Y} = \bigcup_{n \in \mathbb{N}} \prod_{i=1}^{n} \mathcal{V}$ the (countable) space of reasoning steps. Note that mathematically, these two spaces are identical, but we define both $\mathcal{X}$ and $\mathcal{Y}$ for notational convenience. By $\Delta(\mathcal{Y})$, we denote the set of probability measures over $\mathcal{Y}$. For $x \in \mathcal{X}$, let $\pi_B(y \mid x) \in \Delta(\mathcal{Y})$ and $\pi_S(y \mid x) \in \Delta(\mathcal{Y})$ be the *base* and *small* language model distributions over $y \in \mathcal{Y}$ given $x$. Note that we define the distributions over *reasoning steps* instead of single *tokens*. When we write $\pi_B(\cdot \mid x, y)$, it denotes the distribution of $\pi_B$ over $\mathcal{Y}$ given a prompt $x$ and a (partial) response $y$. When $y$ consists of a sequence of reasoning steps $y^t$, we will denote them with superscripts $y = (y^1, ..., y^T)$, $y^t \in \mathcal{Y}$. Note that by definition of $\mathcal{Y}$, we can view $y$ itself as an element of $\mathcal{Y}$ again. Subscripts $y_i$ denote *different samples* generated by the same model.

**Reward Models.** Reward models for LLMs predict how good a generated response $y$ is for a given prompt $x$. They can broadly be split into two classes: *Outcome reward models* (ORMs) assign a reward $r(x, y)$ to a complete response $y$ (i.e. generated until EOS) for a prompt $x \in \mathcal{X}$. *Process reward models* (PRMs) (Lightman et al., 2024) instead assign a reward $r(x, (y^1, ..., y^t))$ to every partial sequence of reasoning steps $(y^1, ..., y^t)$, $t = 1, ..., T$. In the following, we assume we are given a PRM $r : \mathcal{X} \times \mathcal{Y} \to [0, R]$ for some $R < \infty$. We assume that $r$ approximates a *golden reward* (Gao et al., 2023) $r^* : \mathcal{X} \times \mathcal{Y} \to \mathbb{R}_{\geq 0}$, which can be thought of as the "true" reward function.

**Divergences.** Recall that the Kullback–Leibler divergence between two distributions $P, Q \in \Delta(\mathcal{Y})$ with $P \ll Q$ is defined as

$$\mathrm{KL}(P \| Q) = \mathbb{E}_{y \sim P}\left[\log \frac{P(y)}{Q(y)}\right],$$

and the chi-square divergence as

$$\chi^2(P \| Q) = \int \left(\frac{\mathrm{d}P}{\mathrm{d}Q} - 1\right)^2 \mathrm{d}Q = \int \frac{\mathrm{d}P^2}{\mathrm{d}Q} - 1.$$

**KL Regularized Reward Alignment.** A standard formulation for maximizing the reward $r(x, y)$ given $x \in \mathcal{X}$, while constraining how far the policy can move from the base policy $\pi_B(\cdot \mid x)$, is to add a KL regularizer, and find $\pi_B^*$ maximizing

$$\max_{\pi \in \Delta(\mathcal{Y})} \mathbb{E}_{y \sim \pi}[r(x, y)] - \frac{1}{\beta} \mathrm{KL}\left(\pi(\cdot \mid x) \| \pi_B(\cdot \mid x)\right),$$

where $\beta > 0$ trades off maximizing the reward versus fidelity to $\pi_B$. It is well known (e.g. (Korbak et al., 2022)) that the optimal policy has the closed form

$$\pi_{\beta,B}(y \mid x) = \frac{\pi_B(y \mid x) \exp\big(\beta \, r(x,y)\big)}{Z_{\beta,B}(x)}, \tag{1}$$

where $Z_{\beta,B}(x) = \mathbb{E}_{y' \sim \pi_B(\cdot|x)}\left[e^{\beta \, r(x,y')}\right]$. Note that sampling from this distribution becomes intractable when decoding more than one token in $y$ at a time.

**Best-of-$n$ Sampling.** Best-of-$n$ (BoN) (Beirami et al., 2025) is a common inference-time method for scaling LLMs. Best-of-$n$ draws $y_1, \ldots, y_n \sim \pi_B(\cdot \mid x)$, and selects

$$y^* = \arg \max_{i \in \{1,\ldots,n\}} r(x, y_i).$$

Since BoN greedily selects the response that maximizes the reward, it is also sometimes referred to as *hard* best-of-$n$. When the reward model is suboptimal, best-of-$n$ is known to be prone to *reward hacking* (Skalse et al., 2022), which can be mitigated by sampling via *soft best-of-$n$*.

**Soft Best-of-$n$ Sampling.** Soft best-of-$n$ (S-BoN) (Verdun et al., 2025) weighs each drawn sample by a temperature-scaled softmax $w_i \propto \exp\big(\beta \, r(x, y_i)\big)$ (where $\beta$ is an *inverse temperature*), then samples a response $y_i$ with probability $w_i / \sum_j w_j$. We denote the soft best-of-$n$ distribution over $y$ by $\pi_{\beta,B}^n(\cdot \mid x)$. Note that both soft and hard BoN can be applied to one-shot generation (where the complete response it generated in one step) or reasoning tasks, where the $y_i$ correspond to reasoning steps, and the BoN procedure is repeatedly applied. In this work, we focus on reasoning tasks. By moving from hard to soft best-of-$n$, the distribution $\pi_{\beta,B}^n(\cdot \mid x)$ enjoys a KL bound to the tilted distribution $\pi_{\beta,B}$ (Verdun et al., 2025):

$$\mathrm{KL}(\pi_{\beta,B} \| \pi_{\beta,B}^n) \leq \log\left(1 + \frac{\mathrm{Var}_{y \sim \pi_B}[e^{\beta r(x,y)}]}{n(\mathbb{E}_{y \sim \pi_B}[e^{\beta r(x,y)}])^2}\right). \tag{2}$$

In other words, the tilted distribution $\pi_{\beta,B}$ can be approximated by soft best-of-$n$ sampling by letting $n \to \infty$. Aminian et al. (2025) provide a thorough theoretical analysis of soft best-of-$n$ compared to regular best-of-$n$ and show it can mitigate reward hacking.

## 4    GUIDED SPECULATIVE INFERENCE

Our goal is to (approximately) sample from the distribution $\pi_{\beta,B}$. As we have seen, while one cannot sample from the distribution directly, it can be approximated arbitrarily well by soft best-of-$n$ sampling with the target model $\pi_B$, cmp. equation 2, which is linked to the closed-form solution of $\pi_{\beta,B}$ as a reward-tilted version of $\pi_B$ (1). However, this requires autoregressively generating $n$ responses from the target model, which can get prohibitively expensive. We would like to utilize a small draft model $\pi_S$ to accelerate inference, resemblant of speculative decoding. However, the tilted distribution (1) is a distributions over $\pi_B$, not over $\pi_S$. The trick is to note that we can write it as

$$\pi_{\beta,B}(y \mid x) = \frac{\pi_S(y \mid x) \exp\left(\beta r(x,y) + \log\left(\frac{\pi_B(y|x)}{\pi_S(y|x)}\right)\right)}{Z_{\beta,B}(x)},$$

i.e. we can rewrite it as a distribution over $\pi_S$ (exponentially) tilted by the *tilted rewards*

$$\tilde{r}(x,y) = r(x,y) + \frac{1}{\beta} \log\left(\frac{\pi_B(y \mid x)}{\pi_S(y \mid x)}\right),$$

with the convention $\log(0) = -\infty$. This allows us to do soft best-of-$n$ sampling over samples from $\pi_S$ with the tilted rewards $\tilde{r}$ instead of $r$ to approximately sample from $\pi_{\beta,B}$:

**Reward-Likelihood Tilted S-BoN.** For $x \in \mathcal{X}$, the (one-step) reward-tilted S-BoN is defined as:

1. sample $y_1, ..., y_n \sim \pi_S(\cdot \mid x)$
2. compute $\tilde{r}_i = r(x, y_i) + \frac{1}{\beta} \log\left(\frac{\pi_B(y_i|x)}{\pi_S(y_i|x)}\right)$
3. sample $y_i \propto \exp(\beta \tilde{r}_i)$

---

**Algorithm 1** Guided Speculative Inference

---

**Require:** base model $\pi_B$, small model $\pi_S$, PRM $r$, $\beta > 0$, threshold $u \in \mathbb{R}$, $n \in \mathbb{N}$, prompt $x \in \mathcal{X}$

1:  $y \leftarrow ()$                                        #empty response
2:  **for** $t = 0, 1, ...,$ until EOS **do**
3:      Sample $\{y_i^t\}_{i=1}^n \sim \pi_S(\cdot \mid x, y)$ #reasoning steps, generated up to '\n\n'
4:      $\tilde{r}_i \leftarrow r(x, (y, y_i^t)) + \frac{1}{\beta} \left( \log \pi_B(y_i^t \mid x, y) - \log \pi_S(y_i^t \mid x, y) \right), \quad i = 1, ..., n$
5:      Sample index $i^* \sim \text{softmax}(\beta \tilde{r}_1, ..., \beta \tilde{r}_n)$
6:      **if** $\tilde{r}_{i^*} \geq u$ **then**
7:          $y \leftarrow (y, y_{i^*}^t)$                          #append step yˆt_(iˆ*)
8:      **else**
9:          Sample $\{y_j^t\}_{j=1}^n \sim \pi_B(\cdot \mid x, y)$
10:         $r_j \leftarrow r(x, (y, y_j^t)), \quad j = 1, ..., n$
11:         Sample index $j^* \sim \text{softmax}(\beta r_1, ..., \beta r_n)$
12:         $y \leftarrow (y, y_{j^*}^t)$
13:     **end if**
14: **end for**

---

We will denote the distribution generated by this sampling algorithm by $\tilde{\pi}_{\text{GSI}}(\cdot \mid x)$. Crucially, $\log \pi_B(y_i \mid x)$ ***can be efficiently computed in parallel by summing over logprobabilities obtained from a single forward pass,*** as opposed to autoregressive generation in soft best-of-$n$ with $\pi_B$. Of course, we can only hope that $\tilde{\pi}_{\text{GSI}}(\cdot \mid x)$ is close to $\pi_{\beta,B}(\cdot \mid x)$ if the support of $\pi_B$ is sufficiently covered by $\pi_S$, which we make precise with the following uniform coverage assumption, following prior work (Huang et al., 2025). This assumption is reasonable in practice, as any response has non-zero probability when sampling with positive temperature, hence the supremum in Assumption 1 is finite if restricting $\mathcal{Y}$ to responses of some maximal length.

**Assumption 1** (Coverage Assumption). *Throughout, we will assume that*

$$C_\infty(x) := \sup_{y \in \mathcal{Y}: \pi_B(y|x) > 0} \frac{\pi_B(y \mid x)}{\pi_S(y \mid x)} < \infty.$$

Under Assumption 1, reward-likelihood tilted S-BoN with $\pi_S$ indeed approximates the tilted distribution $\pi_{\beta,B}$ in the sense of the following theorem.

**Theorem 1.** *Let $x \in \mathcal{X}$. Assume that the coverage assumption (Assumption 1) holds. Let $u \in \mathbb{R}$ be an acceptance threshold (cmp. Algorithm 1), and $\epsilon > 0$ be an arbitrary accuracy. Assume that*

$$n \geq \frac{\left( \chi^2\left( \pi_B(\cdot \mid x) \,\|\, \pi_S(\cdot \mid x) \right) + 1 \right) e^{2\beta \|r\|_\infty} - 1}{e^\epsilon - 1}.$$

*Then,*

$$\text{KL}\left( \pi_{\beta,B}(\cdot \mid x) \,\|\, \tilde{\pi}_{\text{GSI}}(\cdot \mid x) \right) \leq \epsilon.$$

For a discussion of Theorem 1 and its practical implications, please see Appendix C.5. In addition to sampling from the reward-likelihood tilted S-BoN, we also add a rejection sampling-like threshold on the tilted reward, which triggers resampling from the base model $\pi_B$ in case the tilted reward falls below it. While this is not required for the distributional guarantee from Theorem 1, it improves performance empirically, cmp. Section 5. The complete GSI method can be seen in Algorithm 1. We denote the distribution induced by Algorithm 1 (including the rejection step) as $\pi_{\text{GSI}}$ (i.e. $\pi_{\text{GSI}}$ is equal to $\tilde{\pi}_{\text{GSI}}$ when the sample is accepted, and equal to the soft best-of-$n$ distribution $\pi_{\beta,B}^n$ otherwise).

Note that in principle, it is possible to choose different $n$ for the draft and target models. We leave exploring this for future research. While GSI is, in theory, applicable to one-shot generation tasks, we consider $y^t$ in Algorithm 1 to be a reasoning step, i.e. in each iteration $t$ of the algorithm, drafts are generated until the end of the reasoning step, which is attained when a double line break \n\n is generated. The algorithm generates reasoning steps until an end-of-sequence (EOS) token is created.

In addition to the distributional guarantee from Theorem 1, we can also guarantee that the expected difference in (golden) reward goes to $0$ as $n$ increases.

**Theorem 2** (informal). *Let $x \in \mathcal{X}$. Assume that $\mathbb{E}_{y \sim \pi_{\mathrm{GSI}}(\cdot|x)}[r^*(x,y)] < \infty$ and $\mathbb{E}_{y \sim \pi_{\beta,B}(\cdot|x)}[r^*(x,y)] < \infty$. Furthermore, assume the coverage assumption (Assumption 1) holds. Under mild assumptions on $\pi_{\beta,B}^n$ (Assumption 2 in Appendix A), we have*

$$\mathbb{E}_{y \sim \pi_{\beta,B}(\cdot|x)}[r^*(x,y)] - \mathbb{E}_{y \sim \pi_{\mathrm{GSI}}(\cdot|x)}[r^*(x,y)] \xrightarrow{n \to \infty} 0$$

*at rate $\mathcal{O}(1/\sqrt{n})$.*

Both proofs can be found in Appendix A, where we also provide a formal version of Theorem 2 and an explicit bound in terms of $\frac{1}{\sqrt{n}}$.

## 5 EXPERIMENTS

**Models.** We evaluate GSI on two model families with draft and target models of different sizes, to emphasize that GSI leads to consistent latency gains across families and sizes. For the Qwen2.5-Math family (Qwen Team, 2024), we choose Qwen2.5-Math-1.5B-Instruct as the draft model $\pi_S$, and Qwen2.5-Math-7B-Instruct as target model $\pi_B$. On Qwen3 (Qwen Team, 2025), we choose Qwen3-1.7B as the draft and Qwen3-14B as the target model, and disable thinking mode. We select Qwen2.5-Math-PRM-7B as the PRM $r$ throughout. The rewards lie in $[0, 1]$.

**Implementation.** We implement all models with vLLM (Kwon et al., 2023). The log-likelihoods for $\pi_S$ are computed without any additional computational overhead within the forward pass of $\pi_S$. The log-likelihoods for $\pi_B$ can be computed with minimal computational overhead, as they only require a single forward pass through $\pi_B$. We note that for improved latency gains, verification of draft steps with the PRM and the computation of log-likelihoods of draft steps under $\pi_B$ could be parallelized; however, for simplicity we have not implemented this in our current implementation. Each model is hosted on its own GPU; we evaluated on NVIDIA A100, H100, and H200 GPUs. Our implementation is available at `https://github.com/j-geuter/GSI`.

**Datasets.** We evaluate on the following reasoning benchmarks: MATH500 (Lightman et al., 2024), OlympiadBench (He et al., 2024) (the OE_TO_maths_en_COMP split which is text-only math problems in English), Minerva Math (Lewkowycz et al., 2022), GSM8K (Cobbe et al., 2021), and MMLU-STEM (Hendrycks et al., 2021) (which spans topics such as physics, chemistry, biology, math, astronomy, computer science, and electrical engineering). We decode stepwise with chain-of-thought, where "\n\n" tokens denote the end of a reasoning step; rewards are computed on each reasoning step. Following common practice (Zhang et al., 2024; Cemri et al., 2025; Qiu et al., 2025), we evaluate on randomly selected subsets of the datasets to make evaluation feasible. We select 500 samples per dataset (note that MATH500 contains 500 and Minerva Math 272 samples, hence we use the full datasets). We report $95\%$ confidence intervals on all datasets, computed from evaluations over three different random seeds with $N = 500$ samples each.

**Methods.** We compare GSI against our implementation of RSD (Liao et al., 2025) using the same hyperparameters as in the paper, S-BoN with $\pi_S$, and S-BoN with $\pi_B$. As SPECS (Cemri et al., 2025) does not have a publicly available implementation, we do not compare to it in our experiments. However, we compare to the results reported in their paper in Section 5.1.

Note that we do not compare to vanilla speculative decoding with the draft and target model and stepwise s-BoN sampling (i.e., where $n$ reasoning steps are generated in parallel with vanilla speculative decoding, and then verified with the PRM), since speculative decoding is known to scale very poorly with batch size. Even modern frameworks like EAGLE-2 (Li et al., 2024) have been shown to have a token throughput of up to 50% less than that of the target model at larger batch sizes (Yan et al., 2025). In particular, even sophisticated frameworks like EAGLE-3, that require targeted finetuning of the draft model, have not been evaluated beyond $n = 64$ and do not achieve strictly better throughput than the target model alone (Li et al., 2025). GSI circumvents this issue altogether, as generation both from the draft, as well as the target model remains fully parallelizable.

**Hyperparameters.** We use $\beta = 20$ (see Appendix C.3 for an ablation), $u = 0.5$ (selected empirically amongst a range of values based on accuracy vs. latency trade-off; see Appendix C.4 for an ablation), `temperature` $= 0.7$, and `top_p` $= 1.0$. We set the threshold in RSD to $0.7$, which is the same as in the RSD paper (Liao et al., 2025). Further hyperparameter and implementation details can be found in Appendix B.

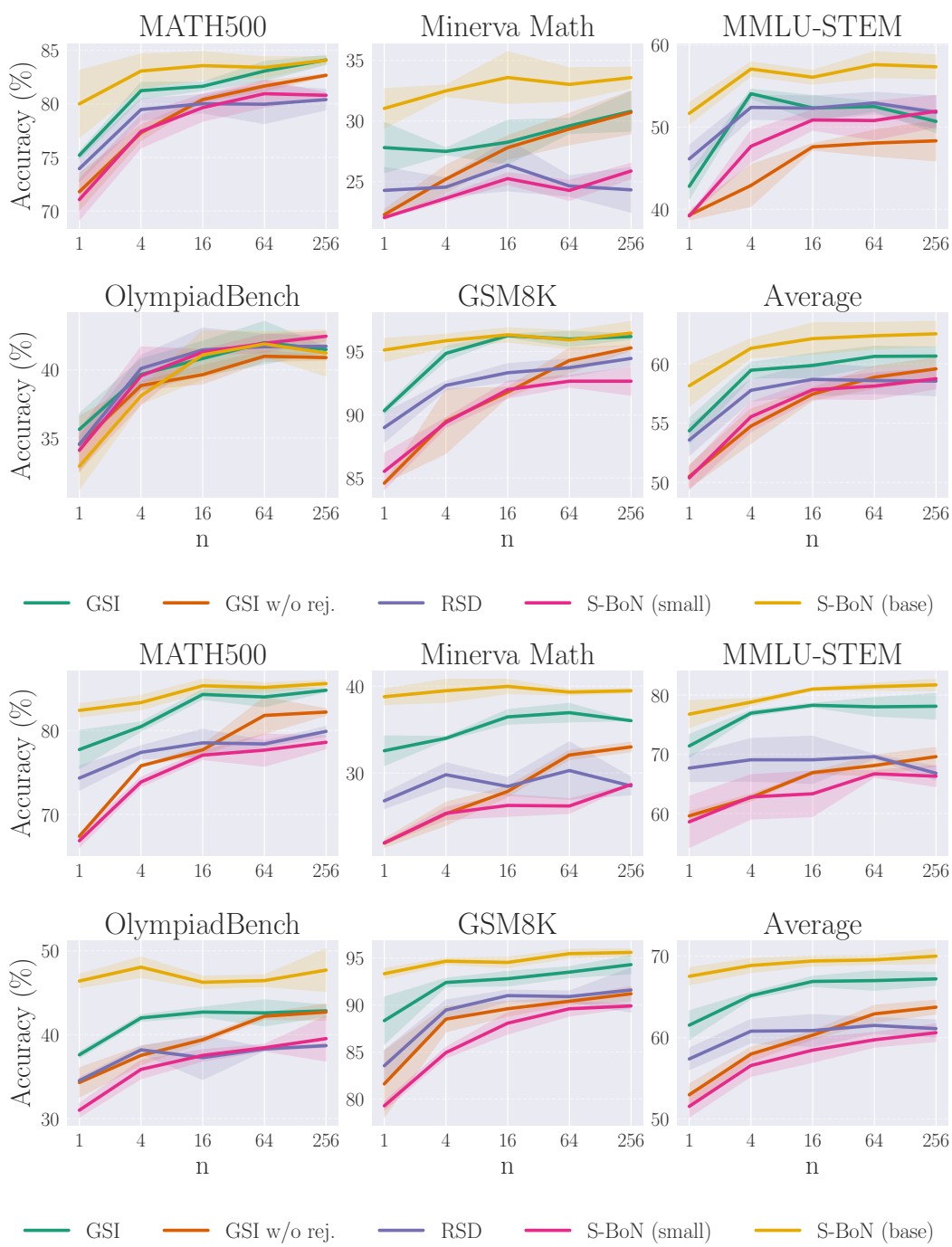

Figure 2: **Qwen2.5-Math (top) / Qwen3 (bottom): GSI outperforms RSD (Liao et al., 2025), soft best-of-$n$ with the draft model, and approaches the performance of soft best-of-$n$ with the base model.** We also compare against GSI without rejection step. The plots contain 95% confidence intervals over three random seeds.

## 5.1 PERFORMANCE ON REASONING BENCHMARKS

Figure 2 compares GSI without the rejection sampling step (i.e., without lines 6 to 11 in Algorithm 1) to regular GSI, S-BoN with $\pi_B$ and $\pi_S$, and RSD. GSI significantly outperforms both soft best-of-$n$ with the draft model, as well as RSD. We see that GSI also clearly outperforms GSI without rejection step; however, this difference becomes less significant as $n$ increases, hinting at the fact that with larger $n$, the samples from the small model reach better coverage of the support of $\pi_B$. Furthermore, on some datasets the accuracy of GSI with and without rejection step approaches or even surpasses the accuracy of $\pi_{\beta,B}^n$, which empirically verifies Theorem 1. We leave investigating this behaviour beyond $n = 256$ for future research. An interesting observation is that amongst all methods, GSI without rejection step seems to benefit most from increasing $n$ and is the only method that does not plateau around $n = 256$. Comparing GSI to SPECS (Cemri et al., 2025) with the accuracies reported in their paper (as there does not exist a public code repository) for the same Qwen2.5-Math models, we see that while SPECS slightly outperforms GSI on OlympiadBench (n = 4: +1.6%, n = 16: +3.2%), GSI is significantly stronger on MATH500 (n = 4: +11.5%, n = 16: +2.9%). Note that we evaluate on subsets of $N = 500$ samples, while SPECS reports accuracies on random subsets of $N = 100$ samples, hence accuracies might not be directly comparable.

In Table 1, we report the inference time per sample (in seconds) across methods (averaged over datasets), as well as the average percentage of samples accepted by GSI and RSD. RSD generally tends to accept almost all samples, which explains why its performance is comparable to S-BoN with the small model (compare Figure 2) while being slightly worse in terms of inference speed. GSI accepts less samples, thus is slower than RSD, while still outperforming S-BoN on the base model in terms of inference speed. For example, on Qwen3 with $n = 16$, GSI achieves a 51% increased throughput in terms of steps per second, with only 3% in relative performance degradation (cmp. Table 3). While GSI tends to generate slightly more steps per problem, this still translates to up to 28% reduced end-to-end latency compared to the target model. An extended version of Table 1 can be found in Appendix C.6. Note that inference times rely on many factors and can be unreliable. For instance, we found that sometimes, vLLM is faster if large batches are artificially fed in sequential chunks instead of one batch. All times reported in Table 1 are for full batches of size $n$. Figure 4 shows how much each of the methods spends on each of the three models, averaged across datasets. We note that our PRM consumes a significant amount of end-to-end runtime, and a more efficient (i.e., smaller) PRM would likely lead to more pronounced latency gains achieved by GSI.

Table 1: **Latency of Qwen2.5 on H100, Qwen3 on A100:** Inference time (in seconds) per reasoning step, number of reasoning steps per sample, acceptance rate, and steps per second (averaged across all datasets, with 95% confidence intervals over three random seeds). GSI is significantly faster than S-BoN on the base model, with up to 51% more steps generated per second.

| Model Family | n | Method | s / step ($\downarrow$) | # steps | % accept | steps / s ($\uparrow$) |
|---|---|---|---|---|---|---|
| Qwen2.5-Math (H100, 1B/7B) | 4 | GSI (ours) | $0.43 \pm 0.03$ | $10.6 \pm 0.3$ | $76.7 \pm 0.1$ | $2.33 \pm 0.15$ |
| | | RSD | $0.34 \pm 0.01$ | $9.7 \pm 0.1$ | $94.9 \pm 0.0$ | $2.94 \pm 0.08$ |
| | | S-BoN (small) | $0.32 \pm 0.01$ | $9.6 \pm 0.0$ | $-$ | $3.12 \pm 0.09$ |
| | | S-BoN (base) | $0.57 \pm 0.01$ | $10.2 \pm 0.3$ | $-$ | $1.75 \pm 0.03$ |
| | 16 | GSI (ours) | $0.72 \pm 0.05$ | $11.4 \pm 0.2$ | $82.0 \pm 0.1$ | $1.39 \pm 0.09$ |
| | | RSD | $0.61 \pm 0.01$ | $10.3 \pm 0.3$ | $97.3 \pm 0.0$ | $1.64 \pm 0.03$ |
| | | S-BoN (small) | $0.52 \pm 0.03$ | $10.3 \pm 0.1$ | $-$ | $1.92 \pm 0.10$ |
| | | S-BoN (base) | $0.94 \pm 0.03$ | $10.5 \pm 0.2$ | $-$ | $1.06 \pm 0.03$ |
| Qwen3 (A100, 1.7B/14B) | 4 | GSI (ours) | $0.56 \pm 0.04$ | $26.7 \pm 0.3$ | $88.0 \pm 0.1$ | $1.79 \pm 0.12$ |
| | | RSD | $0.40 \pm 0.01$ | $28.1 \pm 0.1$ | $97.2 \pm 0.1$ | $2.50 \pm 0.06$ |
| | | S-BoN (s) | $0.38 \pm 0.01$ | $24.8 \pm 0.2$ | $-$ | $2.63 \pm 0.07$ |
| | | S-BoN (b) | $0.83 \pm 0.02$ | $24.9 \pm 0.1$ | $-$ | $1.20 \pm 0.03$ |
| | 16 | GSI (ours) | $1.21 \pm 0.05$ | $27.5 \pm 0.4$ | $91.5 \pm 0.1$ | $0.83 \pm 0.03$ |
| | | RSD | $0.96 \pm 0.01$ | $29.0 \pm 0.2$ | $98.2 \pm 0.1$ | $1.04 \pm 0.01$ |
| | | S-BoN (s) | $1.01 \pm 0.00$ | $26.1 \pm 0.2$ | $-$ | $0.99 \pm 0.00$ |
| | | S-BoN (b) | $1.82 \pm 0.06$ | $24.8 \pm 0.3$ | $-$ | $0.55 \pm 0.02$ |

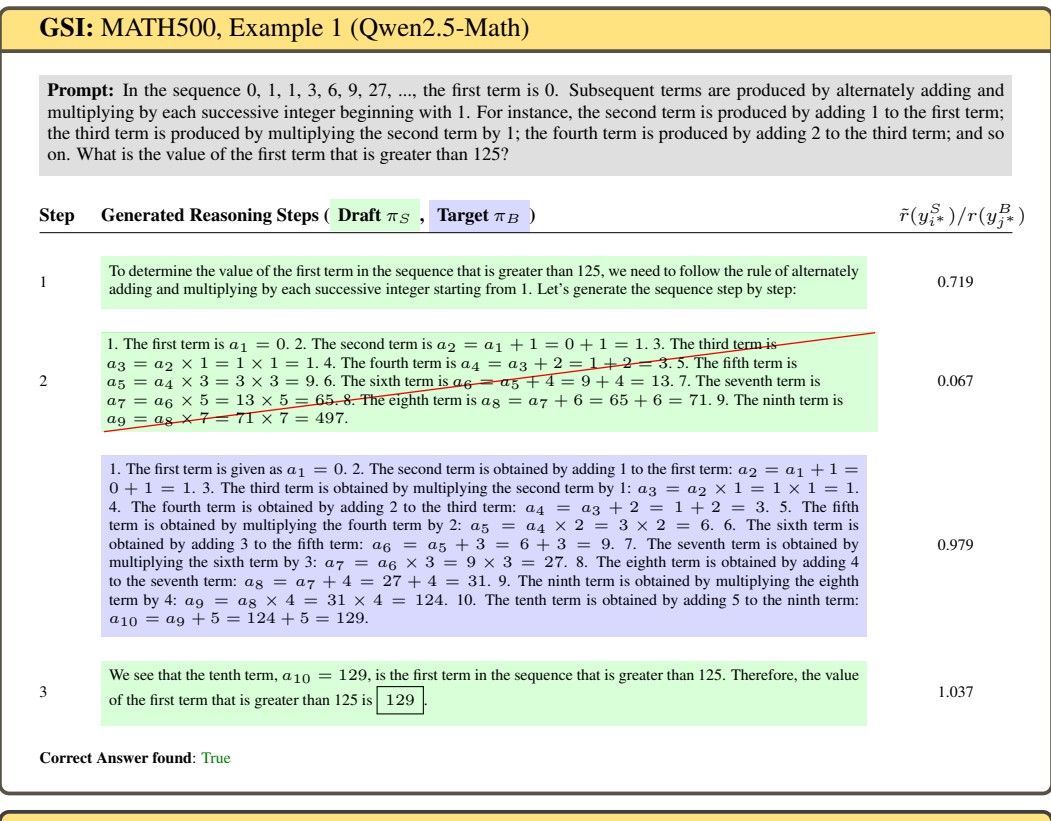

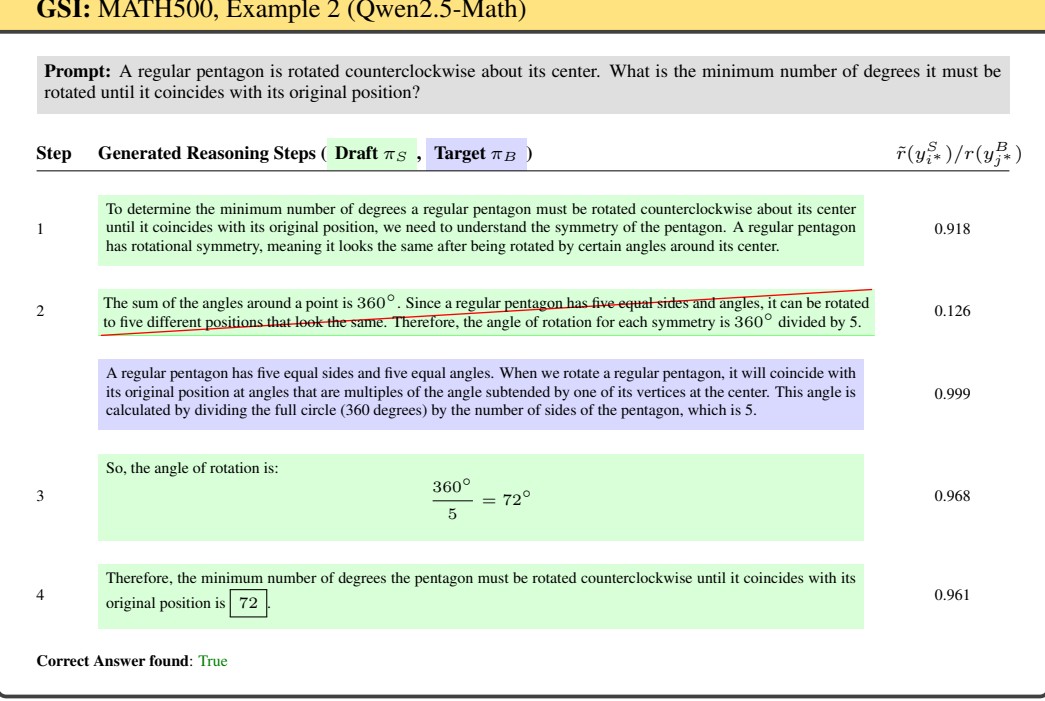

Figure 3: Reasoning traces generated by GSI on two MATH500 samples. **Top:** GSI correctly identifies that the second step generated by the draft model is wrong (crossed out means rejected) and resamples from the base model. **Bottom:** Sometimes, GSI rejects steps that are correct if the base model tends to word them very differently from the draft model.

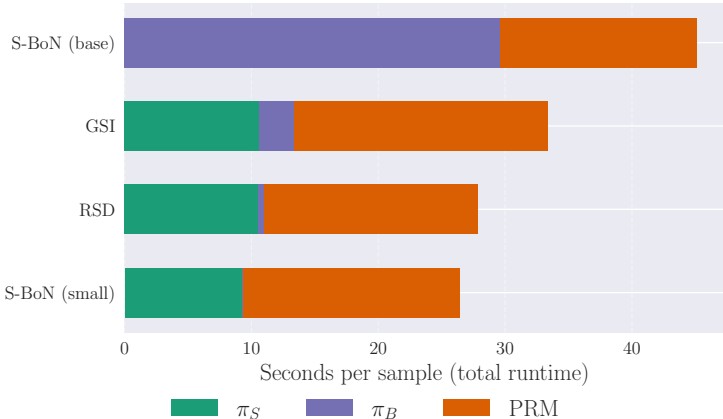

Figure 4: **Qwen3 on A100 runtime breakdown** across $\pi_S$ (1.7B), $\pi_B$ (14B), and the PRM (7B).

## 5.2 GSI REASONING TRACES

Figure 3 shows two sample reasoning traces generated by GSI with the Qwen2.5-Math models for $n = 4$ on MATH500. The last column contains the tilted rewards $\tilde{r}$ for samples from $\pi_S$, and regular rewards $r$ for samples from $\pi_B$, aligning with the GSI algorithm. In the first example, GSI correctly identifies an incorrect step generated by $\pi_S$ in the second step by its small tilted reward, and resamples a correct step from $\pi_B$, whereas the second example shows that tilted rewards $\tilde{r}$ can also sometimes be misleading. We provide additional samples in Appendix C.7, including comparisons to the reasoning traces generated by RSD, and examples that highlight the advantage of using tilted rewards instead of raw rewards.

## 5.3 ABLATIONS

Appendix C contains additional experiments, including more detailed accuracy results, a more detailed comparison of the acceptance rates of GSI and RSD, a discussion of Theorem 1 and its practical implications, and ablations with Qwen2.5-Math on MATH500 over $\beta$ and over $u$, which show that our choice of $\beta = 20$ strikes a balance between weighing $r$ and the log ratio $\log(\pi_B/\pi_S)$, and that the threshold $u = 0.5$ is optimal for smaller values of $n$. Note that while the ablations show that $\beta = 20$ and $u = 0.5$ are sensible choices for MATH500 with Qwen2.5-Math, our experiments confirm that these can be used out-of-the-box across datasets and model families without further hyperparameter search. However, the performance of GSI can likely be improved with a more fine-grained threshold schedule $\{u_n\}_n$ depending on $n$, which we leave for future research.

## 6 DISCUSSION

Developing compute-efficient algorithms remains a critical challenge in test-time scaling of language models. In this work, we introduce *Guided Speculative Inference* (GSI), a novel inference-time algorithm for efficient reward-guided decoding from a target language model. GSI leverages speculative samples from a small draft model to approximate the optimal tilted policy of the target model with respect to a given reward function. We show that unlike previous approaches, GSI provably approaches the optimal policy as the number of samples $n$ generated at each step increases, and can provably achieve expected rewards arbitrarily close to the optimum. Empirical results on various reasoning benchmarks (MATH500, OlympiadBench, Minerva Math, MMLU-STEM, GSM8K), model families (Qwen2.5-Math and Qwen3) and sizes ranging from 1B to 14B parameters show that GSI consistently and significantly outperforms existing approaches, such as reward-guided speculative decoding (Liao et al., 2025), SPECS (Cemri et al., 2025), soft best-of-$n$ with the draft model, and, perhaps surprisingly, even surpasses soft best-of-$n$ with the target model in some cases. Results on inference time show that GSI can efficiently trade off inference time compute for significant performance gains, making it a practical framework for efficient LLM deployment.

REPRODUCIBILITY

The complete code needed to reproduce our results is available at https://github.com/j-geuter/GSI.

ACKNOWLEDGMENTS

We would like to thank Nick Hill and Guangxuan (GX) Xu from Red Hat for their help with vLLM. JG and DAM acknowledge support from the Aramont Fellowship Fund and the FAS Dean's Competitive Fund for Promising Scholarship. JG is supported by a graduate fellowship at the Kempner Institute for the Study of Natural and Artificial Intelligence at Harvard University.

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

CONTENTS

# A PROOFS

## A.1 PROOF OF THEOREM 1

**Theorem 1.** *Let $x \in \mathcal{X}$. Assume that the coverage assumption (Assumption 1) holds. Let $u \in \mathbb{R}$ be an acceptance threshold (cmp. Algorithm 1), and $\epsilon > 0$ be an arbitrary accuracy. Assume that*

$$n \geq \frac{\left(\chi^2\big(\pi_B(\cdot \mid x) \,\|\, \pi_S(\cdot \mid x)\big) + 1\right) e^{2\beta\|r\|_\infty} - 1}{e^\epsilon - 1}.$$

*Then,*

$$\mathrm{KL}\big(\pi_{\beta,B}(\cdot \mid x) \,\|\, \tilde{\pi}_{\mathrm{GSI}}(\cdot \mid x)\big) \ \leq \ \epsilon.$$

*Proof.* By Lemma 1 in (Verdun et al., 2025) (which equally holds for countable spaces $\mathcal{Y}$), we have

$$\tilde{\pi}_{\mathrm{GSI}}(y \mid x) \ \geq \ \frac{\pi_S(y \mid x) \exp\Big[\beta\, r(x,y) + \log \frac{\pi_B(y \mid x)}{\pi_S(y \mid x)}\Big]}{\frac{1}{n} \exp\Big[\beta\, r(x,y) + \log \frac{\pi_B(y|x)}{\pi_S(y|x)}\Big] + \frac{n-1}{n} \mathbb{E}_{y' \sim \pi_S(\cdot|x)}\big[\frac{\pi_B(y'|x)}{\pi_S(y'|x)} e^{\beta\, r(x,y')}\big]}$$

$$= \ \frac{\pi_B(y \mid x)\, e^{\beta\, r(x,y)}}{\frac{1}{n} \frac{\pi_B(y \mid x)}{\pi_S(y \mid x)}\, e^{\beta\, r(x,y)} + \frac{n-1}{n} \mathbb{E}_{y' \sim \pi_B(\cdot|x)}\big[e^{\beta\, r(x,y')}\big]}.$$

Hence

$$\mathrm{KL}\big(\pi_{\beta,B}\|\tilde{\pi}_{\mathrm{GSI}}\big) = \sum_y \pi_{\beta,B}(y \mid x)\, \log \frac{\pi_{\beta,B}(y \mid x)}{\tilde{\pi}_{\mathrm{GSI}}(y \mid x)}$$

$$\leq \sum_y \frac{\pi_B(y \mid x) e^{\beta r(x,y)}}{\mathbb{E}_{y' \sim \pi_B(\cdot|x)}\big[e^{\beta r(x,y')}\big]} \, \log\left( \frac{\pi_B(y \mid x)\, e^{\beta r(x,y)} \left[\frac{1}{n} \frac{\pi_B(y \mid x)}{\pi_S(y \mid x)} e^{\beta r(x,y)} + \frac{n-1}{n}\mathbb{E}_{y' \sim \pi_B(\cdot|x)}\big[e^{\beta r(x,y')}\big]\right]}{\mathbb{E}_{y' \sim \pi_B(\cdot|x)}\big[e^{\beta r(x,y')}\big] \pi_B(y \mid x)\, e^{\beta r(x,y)}} \right)$$

$$= \sum_y \frac{\pi_B(y \mid x) e^{\beta r(x,y)}}{\mathbb{E}_{y' \sim \pi_B(\cdot|x)}\big[e^{\beta r(x,y')}\big]} \, \log\left( \frac{1}{n} \frac{\pi_B(y \mid x)}{\pi_S(y \mid x)} \frac{e^{\beta\, r(x,y)}}{\mathbb{E}_{y' \sim \pi_B(\cdot|x)}\big[e^{\beta\, r(x,y')}\big]} + \frac{n-1}{n} \right)$$

$$\leq \log\left( \frac{1}{n} \left( \sum_y \frac{\pi_B(y \mid x)^2}{\pi_S(y \mid x)} \frac{e^{2\beta\, r(x,y)}}{\big(\mathbb{E}_{y' \sim \pi_B(\cdot|x)}\big[e^{\beta\, r(x,y')}\big]\big)^2} \right) + \frac{n-1}{n} \right) \qquad \text{(Jensen's inequality)}$$

$$\leq \log\left( \frac{1}{n}\, e^{2\beta\,\|r\|_\infty} \frac{\chi^2\big(\pi_B(\cdot \mid x) \,\|\, \pi_S(\cdot \mid x)\big) + 1}{\big(\mathbb{E}_{y' \sim \pi_B(\cdot|x)}\big[e^{\beta\, r(x,y')}\big]\big)^2} + \frac{n-1}{n} \right)$$

$$\leq \log\left( \frac{\big(\chi^2\big(\pi_B(\cdot|x) \,\|\, \pi_S(\cdot|x)\big)+1\big) e^{2\beta\|r\|_\infty}}{n} + \frac{n-1}{n} \right),$$

using the fact that

$$\mathbb{E}_{y' \sim \pi_B(\cdot|x)}\big[e^{\beta\, r(x,y')}\big] \geq 1$$

since $r(x,y') \geq 0$. Now for $\epsilon > 0$, we have

$$\log\left( \frac{\big(\chi^2\big(\pi_B(\cdot|x) \,\|\, \pi_S(\cdot|x)\big)+1\big) e^{2\beta\|r\|_\infty}}{n} + \frac{n-1}{n} \right) \ \leq \ \epsilon$$

$$\Leftrightarrow \qquad \frac{\big(\chi^2\big(\pi_B(\cdot|x) \,\|\, \pi_S(\cdot|x)\big)+1\big) e^{2\beta\|r\|_\infty}}{n} + 1 - \frac{1}{n} \ \leq \ e^\epsilon,$$

$$\Leftrightarrow \qquad 1 + \frac{\big(\chi^2\big(\pi_B(\cdot|x) \,\|\, \pi_S(\cdot|x)\big)+1\big) e^{2\beta\|r\|_\infty} - 1}{n} \ \leq \ e^\epsilon,$$

$$\Leftrightarrow \qquad \frac{\big(\chi^2\big(\pi_B(\cdot|x) \,\|\, \pi_S(\cdot|x)\big)+1\big) e^{2\beta\|r\|_\infty} - 1}{n} \ \leq \ e^\epsilon - 1,$$

$$\Leftrightarrow \qquad \frac{\big(\chi^2\big(\pi_B(\cdot \mid x) \,\|\, \pi_S(\cdot \mid x)\big) + 1\big) e^{2\beta\|r\|_\infty} - 1}{e^\epsilon - 1} \ \leq \ n.$$

$$\qquad\qquad\qquad\qquad\qquad\qquad\qquad\qquad\qquad\qquad\qquad\qquad\qquad\qquad\qquad\qquad \square$$

## A.2 PROOF OF THEOREM 2

**Assumption 2.** *Let $Y_\geq = \{y : \tilde{r}(y) \geq u\}$. Assume that*

$$\frac{1}{\pi^n_{\beta,B}(Y_\geq)} \int_{Y_\geq} r^*(y) \mathrm{d}\pi^n_{\beta,B}(y) \geq \int_Y r^*(y) \mathrm{d}\pi^n_{\beta,B}(y),$$

*in words: $\pi^n_{\beta,B}$ has higher average golden rewards $r^*$ on the set $Y_\geq$ than on the entire set $Y$.*

*Furthermore, assume that*

$$\pi^n_{\beta,B}(Y_\geq) \geq \tilde{\pi}_{\mathrm{GSI}}(Y_\geq),$$

*in words: $\pi^n_{\beta,B}$ is more likely to generate samples with high tilted rewards than $\tilde{\pi}_{\mathrm{GSI}}$.*

**Theorem 2.** *Let $x \in \mathcal{X}$. Assume that $\mathbb{E}_{\pi_{\mathrm{GSI}}}[r^*] < \infty$ and $\mathbb{E}_{\pi_{\beta,B}}[r^*] < \infty$ (here we implicitly assume that distributions and rewards are conditioned on $x$, which we omit for ease of notation). Furthermore, assume that Assumptions 1 and 2 hold. Denote by $p(u)$ the acceptance probability of GSI. Then*

$$\mathbb{E}_{\pi_{\beta,B}}[r^*] - \mathbb{E}_{\pi_{\mathrm{GSI}}}[r^*] \leq \frac{\|r^*\|_\infty}{\sqrt{n}} \left[ p(u)^{\frac{1}{2}} e^{\beta\|r\|_\infty} \left(\chi^2(\pi_B\|\pi_S) + 1\right)^{\frac{1}{2}} + (1-p(u))^{\frac{1}{4}} \left(\mathrm{CV}(e^{\beta r})^2 + 1\right)^{\frac{1}{2}} \right],$$

*where $\mathrm{CV}(e^{\beta r}) := \sqrt{\dfrac{\mathrm{Var}_{y'\sim\pi_B(\cdot|x)}[e^{\beta r(x,y')}]}{\left(\mathbb{E}_{y'\sim\pi_B(\cdot|x)}[e^{\beta r(x,y')}]\right)^2}}$. In particular, we have $\mathbb{E}_{\pi_{\mathrm{GSI}}}[r^*] - \mathbb{E}_{\pi_{\beta,B}}[r^*] \xrightarrow{n\to\infty} 0$ at rate $\mathcal{O}(1/\sqrt{n})$.*

*Proof.* Denote by $Y_\geq \subset Y$ the set where $\tilde{r}(y) \geq u$, and $Y_< = Y \setminus Y_\geq$. Let $x \in \mathcal{X}$. We note that for $y \in Y_\geq$, we have

$$\pi_{\mathrm{GSI}}(y) = \tilde{\pi}_{\mathrm{GSI}}(y) + \tilde{\pi}_{\mathrm{GSI}}(Y_<)\pi^n_{\beta,B}(y),$$

since any $y \in Y_\geq$ can be generated either from $\tilde{\pi}_{\mathrm{GSI}}$ directly (in which case the sample is accepted by the algorithm), or the sample from $\tilde{\pi}_{\mathrm{GSI}}$ is rejected (which happens with probability $\tilde{\pi}_{\mathrm{GSI}}(Y_<)$), in which case $y$ can be generated with $\pi^n_{\beta,B}$. Similarly, for $y \in Y_<$,

$$\pi_{\mathrm{GSI}}(y) = \tilde{\pi}_{\mathrm{GSI}}(Y_<)\pi^n_{\beta,B}(y), \tag{3}$$

as $y \in Y_<$ can only be generated by $\pi_{\mathrm{GSI}}$ if a sample from $\tilde{\pi}_{\mathrm{GSI}}$ is first rejected and then $y$ is generated by $\pi^n_{\beta,B}$. Thus,

$$\mathbb{E}_{\pi_{\beta,B}}[r^*] - \mathbb{E}_{\pi_{\mathrm{GSI}}}[r^*] =$$

$$\mathbb{E}_{\pi_{\beta,B}}[\mathbb{1}_{Y_\geq}r^*] - \mathbb{E}_{\tilde{\pi}_{\mathrm{GSI}}}[\mathbb{1}_{Y_\geq}r^*] - \tilde{\pi}_{\mathrm{GSI}}(Y_<)\mathbb{E}_{\pi^n_{\beta,B}}[\mathbb{1}_{Y_\geq}r^*] + \mathbb{E}_{\pi_{\beta,B}}[\mathbb{1}_{Y_<}r^*] - \tilde{\pi}_{\mathrm{GSI}}(Y_<)\mathbb{E}_{\pi^n_{\beta,B}}[\mathbb{1}_{Y_<}r^*] =$$

$$\mathbb{E}_{\pi_{\beta,B}}[\mathbb{1}_{Y_\geq}r^*] - \mathbb{E}_{\tilde{\pi}_{\mathrm{GSI}}}[\mathbb{1}_{Y_\geq}r^*] + \mathbb{E}_{\pi_{\beta,B}}[\mathbb{1}_{Y_<}r^*] - \mathbb{E}_{\pi^n_{\beta,B}}[\mathbb{1}_{Y_<}r^*] +$$

$$\tilde{\pi}_{\mathrm{GSI}}(Y_\geq)\mathbb{E}_{\pi^n_{\beta,B}}[\mathbb{1}_{Y_<}r^*] - \tilde{\pi}_{\mathrm{GSI}}(Y_<)\mathbb{E}_{\pi^n_{\beta,B}}[\mathbb{1}_{Y_\geq}r^*] =$$

$$\mathbb{E}_{\pi_{\beta,B}}[\mathbb{1}_{Y_\geq}r^*] - \mathbb{E}_{\tilde{\pi}_{\mathrm{GSI}}}[\mathbb{1}_{Y_\geq}r^*] + \mathbb{E}_{\pi_{\beta,B}}[\mathbb{1}_{Y_<}r^*] - \mathbb{E}_{\pi^n_{\beta,B}}[\mathbb{1}_{Y_<}r^*] +$$

$$\tilde{\pi}_{\mathrm{GSI}}(Y_\geq)\mathbb{E}_{\pi^n_{\beta,B}}[\mathbb{1}_{Y_<}r^*] + \tilde{\pi}_{\mathrm{GSI}}(Y_\geq)\mathbb{E}_{\pi^n_{\beta,B}}[\mathbb{1}_{Y_\geq}r^*] - \tilde{\pi}_{\mathrm{GSI}}(Y_<)\mathbb{E}_{\pi^n_{\beta,B}}[\mathbb{1}_{Y_\geq}r^*] - \tilde{\pi}_{\mathrm{GSI}}(Y_\geq)\mathbb{E}_{\pi^n_{\beta,B}}[\mathbb{1}_{Y_\geq}r^*] =$$

$$\underbrace{\mathbb{E}_{\pi_{\beta,B}}[\mathbb{1}_{Y_\geq}r^*] - \mathbb{E}_{\tilde{\pi}_{\mathrm{GSI}}}[\mathbb{1}_{Y_\geq}r^*]}_{(a)} + \underbrace{\mathbb{E}_{\pi_{\beta,B}}[\mathbb{1}_{Y_<}r^*] - \mathbb{E}_{\pi^n_{\beta,B}}[\mathbb{1}_{Y_<}r^*] + \tilde{\pi}_{\mathrm{GSI}}(Y_\geq)\mathbb{E}_{\pi^n_{\beta,B}}[r^*]}_{(b)} \underbrace{- \mathbb{E}_{\pi^n_{\beta,B}}[\mathbb{1}_{Y_\geq}r^*].}_{(c)}$$

**Step 1: Bounding (a).** We have by Cauchy-Schwarz:

$$(a) = \mathbb{E}_{y\sim\pi_{\beta,B}(\cdot|x)}\left[\mathbb{1}_{Y_\geq}(y)\, r^*(x,y)\right] - \mathbb{E}_{y\sim\tilde{\pi}_{\mathrm{GSI}}(\cdot|x)}\left[\mathbb{1}_{Y_\geq}(y)\, r^*(x,y)\right]$$

$$\leq \|r^*\|_\infty \left(\int \mathbb{1}_{Y_\geq}(y)\, \mathrm{d}\tilde{\pi}_{\mathrm{GSI}}(y\mid x)\right)^{\frac{1}{2}} \left(\int \left(\frac{\pi_{\beta,B}(y\mid x) - \tilde{\pi}_{\mathrm{GSI}}(y\mid x)}{\tilde{\pi}_{\mathrm{GSI}}(y\mid x)}\right)^2 \tilde{\pi}_{\mathrm{GSI}}(\mathrm{d}y\mid x)\right)^{\frac{1}{2}}$$

$$= \|r^*\|_\infty \left(\tilde{\pi}_{\mathrm{GSI}}(Y_\geq\mid x)\right)^{\frac{1}{2}} \left(\chi^2(\pi_{\beta,B}(\cdot\mid x)\,\|\,\tilde{\pi}_{\mathrm{GSI}}(\cdot\mid x))\right)^{1/2}. \tag{4}$$

By Lemma 1 from (Verdun et al., 2025) we have

$$
\chi^2\big(\pi_{\beta,B}(\cdot \mid x) \,\|\, \tilde{\pi}_{\mathrm{GSI}}(\cdot \mid x)\big) \tag{5}
$$

$$
= \int \frac{\pi_{\beta,B}(y \mid x)^2}{\tilde{\pi}_{\mathrm{GSI}}(y \mid x)} \, \mathrm{d}y \; - \; 1
$$

$$
= \int \frac{\big(\pi_B(y \mid x)\, e^{\beta\, r(x,y)}\big)^2}{\big(\mathbb{E}_{y'\sim\pi_B(\cdot|x)}[e^{\beta r(x,y')}]\big)^2 \, \tilde{\pi}_{\mathrm{GSI}}(y \mid x)} \, \mathrm{d}y \; - \; 1
$$

$$
\overset{\text{Lemma 1}}{\leq} \int \frac{\big(\pi_B(y \mid x)\, e^{\beta r(x,y)}\big)^2}{\big(\mathbb{E}_{y'\sim\pi_B(\cdot|x)}[e^{\beta r(x,y')}]\big)^2} \, \frac{\frac{1}{n}\frac{\pi_B(y|x)}{\pi_S(y|x)}e^{\beta r(x,y)} + \frac{n-1}{n}\,\mathbb{E}_{y'\sim\pi_B(\cdot|x)}[e^{\beta r(x,y')}]}{\pi_B(y \mid x)\, e^{\beta r(x,y)}} \, \mathrm{d}y \; - \; 1
$$

$$
= \frac{1}{n\,\big(\mathbb{E}_{y'\sim\pi_B(\cdot|x)}[e^{\beta r(x,y')}]\big)^2} \int \frac{\pi_B(y \mid x)^2}{\pi_S(y \mid x)}\, e^{2\beta r} \, \mathrm{d}y + \frac{n-1}{n} \; - \; 1
$$

$$
\leq \frac{e^{2\beta\|r\|_\infty}}{n\,\big(\mathbb{E}_{y'\sim\pi_B(\cdot|x)}[e^{\beta r(x,y')}]\big)^2} \Big(\chi^2\big(\pi_B(\cdot \mid x)\|\pi_S(\cdot \mid x)\big) + 1\Big) \; - \; \frac{1}{n}
$$

$$
\leq \frac{1}{n}\, e^{2\beta\|r\|_\infty} \Big(\chi^2\big(\pi_B(\cdot \mid x)\|\pi_S(\cdot \mid x)\big) + 1\Big). \tag{6}
$$

Plugging equation 6 into equation 4 yields

$$
\text{(a)} \;\leq\; \|r^*\|_\infty\; (\tilde{\pi}_{\mathrm{GSI}}(Y_\geq \mid x))^{\frac{1}{2}} \left(\frac{1}{n}\, e^{2\beta\|r\|_\infty} \big(\chi^2(\pi_B\|\pi_S) + 1\big)\right)^{\frac{1}{2}}
$$

$$
= \; \frac{\|r^*\|_\infty}{\sqrt{n}}\, p(u)^{\frac{1}{2}} e^{\beta\|r\|_\infty} \big(\chi^2(\pi_B\|\pi_S) + 1\big)^{\frac{1}{2}}. \tag{7}
$$

**Step 2: Bounding (b).** Similar to the bound for (a), we get

$$
\text{(b)} = \int \mathbb{1}_{Y_<}(y) r^*(x,y) \frac{\pi_{\beta,B}(y \mid x) - \pi_{\beta,B}^n(y \mid x)}{\pi_{\beta,B}^n(y \mid x)} \pi_{\beta,B}^n(\mathrm{d}y \mid x)
$$

$$
\leq \left(\int \mathbb{1}_{Y<}(y) r^*(x,y)^2 \pi_{\beta,B}^n(\mathrm{d}y \mid x)\right)^{\frac{1}{2}} \left(\int \left(\frac{\pi_{\beta,B}(y \mid x) - \pi_{\beta,B}^n(y \mid x)}{\pi_{\beta,B}^n(y \mid x)}\right)^2 \pi_{\beta,B}^n(\mathrm{d}y \mid x)\right)^{\frac{1}{2}}
$$

$$
\leq \pi_{\beta,B}^n(\mathbb{1}_{Y<})^{\frac{1}{2}} \|r^*\|_\infty \left(\int \left(\frac{\pi_{\beta,B}(y \mid x) - \pi_{\beta,B}^n(y \mid x)}{\pi_{\beta,B}^n(y \mid x)}\right)^2 \pi_{\beta,B}^n(\mathrm{d}y \mid x)\right)^{\frac{1}{2}}
$$

$$
< (1 - p(u))^{\frac{1}{4}} \|r^*\|_\infty \big(\chi^2(\pi_{\beta,B}\|\pi_{\beta,B}^n)\big)^{\frac{1}{2}} \tag{8}
$$

by Cauchy-Schwarz. In the last step, we used the second part of Assumption 2, which implies $\pi_{\beta,B}^n(Y_<) < \tilde{\pi}_{\mathrm{GSI}}(Y_<)$, hence

$$
\pi_{\beta,B}^n(Y_<)^{\frac{1}{2}} < \tilde{\pi}_{\mathrm{GSI}}(Y_<)^{\frac{1}{4}} \pi_{\beta,B}^n(Y_<)^{\frac{1}{4}} = \pi_{\mathrm{GSI}}(Y_<)^{\frac{1}{4}}
$$

by equation 3.

Again, using Lemma 1 from (Verdun et al., 2025) we get

$$
\begin{aligned}
\chi^2(\pi_{\beta,B}||\pi^n_{\beta,B}) &= \int \frac{\pi_{\beta,B}(y\mid x)^2}{\pi^n_{\beta,B}(y\mid x)}\, dy \;-\; 1 \\
&\overset{\text{Lemma 1}}{\leq} \int \frac{\big(\pi_B(y\mid x)\, e^{\beta r(x,y)}\big)^2}{\big(\mathbb{E}_{y'\sim\pi_B(\cdot\mid x)}[e^{\beta r(x,y')}]\big)^2}\; \frac{\frac{1}{n}e^{\beta r(x,y)} + \frac{n-1}{n}\mathbb{E}_{y'\sim\pi_B(\cdot\mid x)}[e^{\beta r(x,y')}]}{\pi_B(y\mid x)\, e^{\beta r(x,y)}}\, dy \;-\; 1 \\
&= \frac{1}{n}\frac{\mathbb{E}_{y'\sim\pi_B(\cdot\mid x)}[e^{2\beta r(x,y')}]}{\big(\mathbb{E}_{y'\sim\pi_B(\cdot\mid x)}[e^{\beta r(x,y')}]\big)^2} + \frac{n-1}{n} - 1 \\
&\leq \frac{1}{n}\frac{\mathbb{E}_{y'\sim\pi_B(\cdot\mid x)}[e^{2\beta r(x,y')}]}{\big(\mathbb{E}_{y'\sim\pi_B(\cdot\mid x)}[e^{\beta r(x,y')}]\big)^2} \\
&= \frac{1}{n}\left(\frac{\mathrm{Var}_{y'\sim\pi_B(\cdot\mid x)}[e^{\beta r(x,y')}]}{\big(\mathbb{E}_{y'\sim\pi_B(\cdot\mid x)}[e^{\beta r(x,y')}]\big)^2} + 1\right).
\end{aligned}
\tag{9}
$$

Plugging equation 9 into equation 8 yields

$$
(b) \leq \frac{\|r^*\|_\infty}{\sqrt{n}}(1-p(u))^{\frac{1}{4}}\left(\frac{\mathrm{Var}_{y'\sim\pi_B(\cdot\mid x)}[e^{\beta r(x,y')}]}{\big(\mathbb{E}_{y'\sim\pi_B(\cdot\mid x)}[e^{\beta r(x,y')}]\big)^2} + 1\right)^{\frac{1}{2}}
\tag{10}
$$

**Step 3: Bounding (c).** We have by Assumption 2:

$$
\begin{aligned}
(c) &= \tilde{\pi}_{\mathrm{GSI}}(Y_\geq)\mathbb{E}_{\pi^n_{\beta,B}}[r^*] - \mathbb{E}_{\pi^n_{\beta,B}}[\mathbb{1}_{Y_\geq}r^*] \\
&\leq \frac{\tilde{\pi}_{\mathrm{GSI}}(Y_\geq)}{\pi^n_{\beta,B}(Y_\geq)}\mathbb{E}_{\pi^n_{\beta,B}}[\mathbb{1}_{Y_\geq}r^*] - \mathbb{E}_{\pi^n_{\beta,B}}[\mathbb{1}_{Y_\geq}r^*] \\
&\leq 0,
\end{aligned}
\tag{11}
$$

where we use the first part of Assumption 2 in the first inequality, and the second part in the second inequality.

Combining equations (7), (10), and (11) gives

$$
\mathbb{E}_{\pi_{\beta,B}}[r^*] - \mathbb{E}_{\pi_{\mathrm{GSI}}}[r^*] \leq \frac{\|r^*\|_\infty}{\sqrt{n}}\left[p(u)^{\frac{1}{2}}e^{\beta\|r\|_\infty}\big(\chi^2(\pi_B\|\pi_S)+1\big)^{\frac{1}{2}} + (1-p(u))^{\frac{1}{4}}\big(\mathrm{CV}(e^{\beta r})^2+1\big)^{\frac{1}{2}}\right]
$$

as desired.

$\square$

*Remark* 3. Asymptotically, we also get

$$
\mathbb{E}_{\pi_{\mathrm{GSI}}}[r^*] - \mathbb{E}_{\pi_{\beta,B}}[r^*] \xrightarrow{n\to\infty} 0
$$

without assuming the second part in Assumption 2, since by Theorem 1 combined with Lemma 1 from Verdun et al. (2025), we get

$$
\mathrm{KL}(\pi_{\beta,B}||\tilde{\pi}_{\mathrm{GSI}}) \xrightarrow{n\to\infty} 0 \quad \text{and} \quad \mathrm{KL}(\pi_{\beta,B}||\pi^n_{\beta,B}) \xrightarrow{n\to\infty} 0,
$$

which implies

$$
\frac{\tilde{\pi}_{\mathrm{GSI}}(Y_\geq)}{\pi^n_{\beta,B}(Y_\geq)} \xrightarrow{n\to\infty} 1,
$$

assuming that $\pi_{\beta,B}(Y_\geq) > 0$. This means that term (c) in the proof of Theorem 2 converges to 0 (not necessarily from below). Furthermore, one could simply bound the term $\pi^n_{\beta,B}(\mathbb{1}_{Y<})^{\frac{1}{2}}$ appearing in our derivation of the bound on term (b) by 1, which avoids relying on the second part of Assumption 2 for term (b) and guarantees the asymptotic behaviour above.

## B IMPLEMENTATION DETAILS

This section contains additional implementation details.

### B.1 SYSTEM PROMPTS

We slightly adapt system prompts based on the dataset. Our base system prompt is:

```
"Please reason step by step, and put your final answer within
\\boxed{}."
```

On Minerva, we append it by

```
Do not include units in your final answer.  For example, if the
answer is '5 m/s', write '\\boxed{5}'."
```

On MMLU, we instead use

```
"Please reason step by step, and select the answer from the
given choices 1, 2, 3, or 4.  Respond only with the number of the
correct answer, from 1 to 4, not with the answer itself.  Put the
index of the correct answer within \\boxed{}."
```

### B.2 GENERATION DETAILS

We set `max_new_tokens` in vLLM to 512 (this is the maximum number of tokens per reasoning step). If the rewards of all draft steps lie below $0.1$, we stop generation for that sample and count it as "solved incorrectly", as we have observed that such generations lead to incorrect solutions. On Qwen2.5-Math models, we had used a maximum number of reasoning steps of $45$ (after $45$ steps without finding a solution, the sample counts as "solved incorrectly"). However, as Qwen3 models tend to generate significantly larger numbers of reasoning steps, we increased this limit to $100$ on Qwen3. We use a maximum context window of $8192$ for all three models (if a response exceeds this context size, it counts as "solved incorrectly").

## C ADDITIONAL EXPERIMENTS

### C.1 EXTENDED ACCURACY RESULTS

In Tables 2 and 3, we report the average accuracies of GSI, RSD, S-BoN with $\pi_S$, and S-BoN with $\pi_B$, for both model families. While GSI outperforms RSD and S-BoN with $\pi_S$ on both model families, this difference is more significant with the Qwen3 models, since the Qwen3 draft and target model exhibit a larger performance difference than our Qwen2.5-Math models.

### C.2 ACCEPTANCE RATIOS

In Figure 5, we plot the average acceptance ratio of GSI and RSD for both Qwen2.5-Math and Qwen3. The acceptance ratio of GSI increases from an average $70\%$ (Qwen2.5-Math) and $80\%$ (Qwen3) to around $90\%$ at $n = 256$. That of RSD is significantly higher, increasing from $90\%$ (Qwen2.5-Math) resp. $95\%$ (Qwen3) to almost $100\%$ ($n = 256$), suggesting that as $n$ increases, RSD collapses to soft best-of-$n$ with $\pi_S$, at least without more careful hyperparameter tuning. We note that more intricate acceptance threshold schedules could stabilize acceptance rates across $n$, compare Section C.4. We leave exploring such approaches for future research.

### C.3 ABLATION OVER $\beta$

GSI relies on temperature-scaled soft best-of-$n$ sampling with parameter $\beta$ (corresponding to an inverse temperature) both for samples from $\pi_S$, as well as for samples from $\pi_B$ in case the draft sample gets rejected. Increasing $\beta$ leads to convergence to greedy best-of-$n$, while reducing it converges to random choice. To better understand the behaviour of GSI in terms of $\beta$, we evaluate GSI with Qwen2.5-Math on MATH500 for different values of $\beta$, ranging from $\beta = 0$ (i.e. ignoring $r$,

Table 2: **Qwen2.5-Math:** Accuracy on reasoning benchmarks (95% confidence intervals over three random seeds). GSI consistently outperforms RSD (Liao et al., 2025) and soft best-of-n (S-BoN) with the small model. S-BoN with the base model represents the target distribution. On average, GSI surpasses all baselines and closely approaches the performance of the base-model S-BoN. As $n$ grows, performance saturates.

| n | Method | MATH500 | OlympiadBench | Minerva | MMLU | GSM8K | Average |
|---|---|---|---|---|---|---|---|
| 1 | GSI (ours) | **75.3** $\pm$ 0.5 | **35.6** $\pm$ 1.2 | **27.8** $\pm$ 2.1 | 42.8 $\pm$ 1.6 | **90.3** $\pm$ 0.3 | **54.4** $\pm$ 1.1 |
| | RSD | 74.0 $\pm$ 1.2 | 34.5 $\pm$ 0.6 | 24.3 $\pm$ 1.9 | **46.1** $\pm$ 1.7 | 89.0 $\pm$ 1.2 | 53.6 $\pm$ 1.3 |
| | S-BoN (s) | 71.1 $\pm$ 1.9 | 34.1 $\pm$ 1.5 | 22.1 $\pm$ 0.1 | 39.2 $\pm$ 0.0 | 85.5 $\pm$ 1.4 | 50.4 $\pm$ 1.0 |
| | S-BoN (b) | 80.0 $\pm$ 3.1 | 32.9 $\pm$ 1.7 | 31.0 $\pm$ 1.6 | 51.7 $\pm$ 1.3 | 95.1 $\pm$ 0.9 | 58.2 $\pm$ 1.7 |
| 4 | GSI (ours) | **81.2** $\pm$ 0.9 | 39.7 $\pm$ 1.1 | **27.5** $\pm$ 0.3 | **54.1** $\pm$ 0.6 | **94.9** $\pm$ 0.6 | **59.5** $\pm$ 0.7 |
| | RSD | 79.5 $\pm$ 1.0 | **40.1** $\pm$ 1.0 | 24.5 $\pm$ 0.8 | 52.4 $\pm$ 1.5 | 92.3 $\pm$ 0.3 | 57.8 $\pm$ 0.9 |
| | S-BoN (s) | 77.4 $\pm$ 1.5 | 39.6 $\pm$ 2.1 | 23.6 $\pm$ 0.3 | 47.7 $\pm$ 1.9 | 89.4 $\pm$ 0.4 | 55.5 $\pm$ 1.2 |
| | S-BoN (b) | 83.1 $\pm$ 1.6 | 38.1 $\pm$ 0.6 | 32.5 $\pm$ 0.5 | 57.1 $\pm$ 0.9 | 95.9 $\pm$ 0.5 | 61.3 $\pm$ 0.8 |
| 16 | GSI (ours) | **82.2** $\pm$ 0.6 | 40.8 $\pm$ 1.3 | **28.2** $\pm$ 1.8 | 52.3 $\pm$ 1.5 | **96.3** $\pm$ 0.1 | **60.0** $\pm$ 1.1 |
| | RSD | 80.0 $\pm$ 0.9 | **41.5** $\pm$ 1.6 | 26.4 $\pm$ 2.1 | 52.3 $\pm$ 1.5 | 93.3 $\pm$ 0.7 | 58.7 $\pm$ 1.4 |
| | S-BoN (s) | 79.7 $\pm$ 1.3 | 41.3 $\pm$ 0.2 | 25.2 $\pm$ 0.5 | 50.9 $\pm$ 1.2 | 92.0 $\pm$ 0.7 | 57.8 $\pm$ 0.8 |
| | S-BoN (b) | 83.6 $\pm$ 1.3 | 41.1 $\pm$ 1.8 | 33.6 $\pm$ 2.1 | 56.1 $\pm$ 0.9 | 96.3 $\pm$ 0.6 | 62.1 $\pm$ 1.3 |
| 64 | GSI (ours) | **83.4** $\pm$ 0.5 | **42.0** $\pm$ 1.6 | **29.6** $\pm$ 0.6 | 52.5 $\pm$ 0.6 | **96.0** $\pm$ 0.6 | **60.7** $\pm$ 0.8 |
| | RSD | 80.0 $\pm$ 1.8 | 41.7 $\pm$ 0.9 | 24.6 $\pm$ 0.9 | **52.9** $\pm$ 1.3 | 93.7 $\pm$ 0.7 | 58.6 $\pm$ 1.1 |
| | S-BoN (s) | 80.9 $\pm$ 1.3 | **42.0** $\pm$ 0.7 | 24.3 $\pm$ 0.8 | 50.8 $\pm$ 2.4 | 92.7 $\pm$ 0.5 | 58.1 $\pm$ 1.1 |
| | S-BoN (b) | 83.4 $\pm$ 1.0 | 41.9 $\pm$ 0.9 | 33.0 $\pm$ 1.4 | 57.6 $\pm$ 1.6 | 95.9 $\pm$ 0.7 | 62.4 $\pm$ 1.1 |
| 256 | GSI (ours) | **84.1** $\pm$ 0.4 | 41.5 $\pm$ 0.2 | **30.8** $\pm$ 1.6 | 50.7 $\pm$ 1.4 | **96.2** $\pm$ 0.2 | **60.7** $\pm$ 0.8 |
| | RSD | 80.4 $\pm$ 1.0 | 41.7 $\pm$ 0.8 | 24.3 $\pm$ 1.8 | 51.8 $\pm$ 2.0 | 94.5 $\pm$ 0.6 | 58.5 $\pm$ 1.2 |
| | S-BoN (s) | 80.8 $\pm$ 0.2 | **42.5** $\pm$ 0.3 | 25.9 $\pm$ 0.6 | **51.9** $\pm$ 1.9 | 92.7 $\pm$ 1.1 | 58.7 $\pm$ 0.9 |
| | S-BoN (b) | 84.1 $\pm$ 0.3 | 41.2 $\pm$ 1.7 | 33.6 $\pm$ 0.9 | 57.3 $\pm$ 1.5 | 96.5 $\pm$ 0.9 | 62.5 $\pm$ 1.1 |

and sampling from softmax$(\beta \tilde{r}_i) = $ softmax$(\log(\pi_B(y_i)/\pi_S(y_i)))$ to $\beta = 1000$. Figure 6 depicts the average acceptance ratio of GSI on MATH500 for different values of $\beta$. A sharp phase transition between $\beta = 8$ and $\beta = 20$ can be observed. In Figure 7 we plot the average accuracy for different values of $\beta$ on MATH500, in terms of both $n$ and seconds per reasoning step. While $\beta = 20$ is not uniformly better than other values, it achieves best accuracy overall. These figures demonstrate that $\beta = 20$ strikes a balance in weighing the raw reward $r$ and the log ratio $\log(\pi_B/\pi_S)$, leading to acceptance ratios that are neither too low nor too high and good accuracies overall.

## C.4 ABLATION OVER $u$

A crucial hyperparameter in GSI is the acceptance threshold $u$, compare Algorithm 1. To better understand the behaviour of GSI with respect to $u$, we plot the average acceptance ratios of GSI with Qwen2.5-Math on MATH500 for different values of $u$ in Figure 9, and the average accuracy (over $n$ and over seconds per reasoning step) in Figure 10. As is to be expected, higher thresholds $u$ tend to have lower acceptance rates and higher accuracies, as they sample from the target model $\pi_B$ more frequently. Hence, it is important to choose $u$ in such a way that it strikes a balance between accuracy and latency. In Figure 11 we show an empirical Pareto frontier of $u$ as a function of $n$. This suggests that the optimal $u$ depends on $n$, and an adaptive threshold schedule $\{u_n\}_n$ could improve GSI in terms of accuracy-vs-latency trade-off. For simplicity, we pick a constant value $u = 0.5$ and leave exploring more intricate choices for future research.

## C.5 DISCUSSION OF THEOREM 1

We provide a short discussion of Theorem 1 and its practical implications. Note that while Assumption 1 is necessary in order for all objects in the proof of Theorem 1 to be well-defined, it does not directly impact the bound appearing in Theorem 1. The important quantity here is the chi-squared

Table 3: **Qwen3:** Accuracies on reasoning benchmarks with 95% confidence intervals. GSI outperforms RSD (Liao et al., 2025) and S-BoN with the small model much more significantly than on Qwen-2.5-Math. As $n$ grows, performance saturates.

| n | Method | MATH500 | OlympiadBench | Minerva | MMLU | GSM8K | Average |
|---|--------|---------|---------------|---------|------|-------|---------|
| 1 | GSI (ours) | **77.7** ± 2.3 | **37.6** ± 0.4 | **32.6** ± 1.7 | **71.4** ± 2.0 | **88.3** ± 2.5 | **61.5** ± 1.8 |
|   | RSD | 74.3 ± 1.5 | 34.5 ± 0.3 | 26.8 ± 0.9 | 67.7 ± 2.3 | 83.5 ± 1.7 | 57.4 ± 1.3 |
|   | S-BoN (s) | 66.9 ± 0.8 | 31.0 ± 0.8 | 22.0 ± 0.2 | 58.6 ± 4.3 | 79.3 ± 0.5 | 51.5 ± 1.3 |
|   | S-BoN (b) | 82.4 ± 0.8 | 46.4 ± 0.8 | 38.8 ± 0.9 | 76.8 ± 2.3 | 93.3 ± 0.6 | 67.5 ± 1.1 |
| 4 | GSI (ours) | **80.5** ± 0.6 | **42.0** ± 0.4 | **34.0** ± 0.2 | **76.9** ± 0.5 | **92.4** ± 0.5 | **65.2** ± 0.4 |
|   | RSD | 77.4 ± 0.8 | 38.2 ± 0.5 | 29.8 ± 1.4 | 69.1 ± 3.6 | 89.5 ± 1.1 | 60.8 ± 1.5 |
|   | S-BoN (s) | 73.9 ± 0.6 | 35.9 ± 1.1 | 25.4 ± 0.7 | 62.8 ± 3.7 | 84.9 ± 0.6 | 56.6 ± 1.3 |
|   | S-BoN (b) | 83.3 ± 0.9 | 48.0 ± 1.2 | 39.5 ± 1.3 | 78.8 ± 0.6 | 94.7 ± 0.3 | 68.9 ± 0.9 |
| 16 | GSI (ours) | **84.3** ± 0.6 | **42.7** ± 0.6 | **36.5** ± 0.8 | **78.3** ± 0.3 | **92.8** ± 0.8 | **66.9** ± 0.6 |
|   | RSD | 78.5 ± 1.6 | 37.3 ± 2.6 | 28.5 ± 1.0 | 69.1 ± 4.0 | 91.0 ± 0.6 | 60.9 ± 2.0 |
|   | S-BoN (s) | 77.1 ± 0.5 | 37.5 ± 0.7 | 26.3 ± 1.3 | 63.3 ± 3.9 | 88.1 ± 1.2 | 58.5 ± 1.5 |
|   | S-BoN (b) | 85.3 ± 0.8 | 46.2 ± 0.7 | 40.0 ± 0.8 | 81.0 ± 0.2 | 94.5 ± 0.6 | 69.4 ± 0.6 |
| 64 | GSI (ours) | **84.0** ± 1.2 | **42.6** ± 1.6 | **36.9** ± 1.1 | **78.0** ± 1.6 | **93.5** ± 0.6 | **67.0** ± 1.2 |
|   | RSD | 78.4 ± 0.4 | 38.3 ± 0.2 | 30.3 ± 3.3 | 69.6 ± 0.4 | 90.9 ± 0.6 | 61.5 ± 1.0 |
|   | S-BoN (s) | 77.7 ± 1.9 | 38.5 ± 0.4 | 26.2 ± 0.8 | 66.7 ± 0.6 | 89.6 ± 0.8 | 59.7 ± 0.9 |
|   | S-BoN (b) | 85.1 ± 0.6 | 46.5 ± 0.7 | 39.3 ± 0.3 | 81.4 ± 0.6 | 95.5 ± 0.5 | 69.6 ± 0.5 |
| 256 | GSI (ours) | **84.8** ± 0.2 | **42.9** ± 0.7 | **36.0** ± 0.0 | **78.1** ± 2.2 | **94.3** ± 1.0 | **67.2** ± 0.8 |
|   | RSD | 79.9 ± 0.6 | 38.7 ± 0.2 | 28.5 ± 1.0 | 66.8 ± 1.2 | 91.6 ± 2.4 | 61.1 ± 1.1 |
|   | S-BoN (s) | 78.6 ± 0.8 | 39.5 ± 2.7 | 28.7 ± 0.0 | 66.3 ± 1.8 | 89.9 ± 0.2 | 60.6 ± 1.1 |
|   | S-BoN (b) | 85.6 ± 0.4 | 47.7 ± 2.5 | 39.5 ± 0.3 | 81.7 ± 1.0 | 95.6 ± 0.4 | 70.0 ± 0.9 |

divergence $\chi^2(\pi_B(\cdot \mid x) \| \pi_S(\cdot \mid x))$ between $\pi_B(\cdot \mid x)$ and $\pi_S(\cdot \mid x)$, where $x$ corresponds to either the prompt, or the prompt concatenated with all reasoning steps generated up to a certain point. In Table 4, we show that the chi-squared divergence is generally well-behaved in practice, with mean values of between 1.48 and 3.91, depending on the model family. These values are Monte Carlo estimates over 50 samples from MATH500, where we average both over samples, as well as over reasoning steps in the generation. In each step $t$, we generate $N = 64$ subsequent reasoning steps $y_1^t, ..., y_{64}^t \sim \pi_S(\cdot \mid (x, y^1, ..., y^{t-1}))$, and estimate the $\chi^2$ for that step as

$$\frac{1}{N} \sum_{i=1}^{N} \left( \frac{\pi_B(y_i^t \mid (x, y^1, ..., y^{t-1}))}{\pi_S(y_i^t \mid (x, y^1, ..., y^{t-1}))} - 1 \right)^2 =$$

$$\frac{1}{N} \sum_{i=1}^{N} \left( \exp\left( \log \pi_B(y_i^t \mid (x, y^1, ..., y^{t-1})) - \pi_S(y_i^t \mid (x, y^1, ..., y^{t-1})) \right) - 1 \right)^2$$

from the logprobabilities computed under both models.

Table 4: **Empirical estimates of** $\chi^2(\pi_B(\cdot \mid x) \| \pi_S(\cdot \mid x))$. Averaged over 50 samples from MATH500 and reasoning steps. Monte Carlo estimates with $n = 64$ samples in each step.

| Model Family | mean $\chi^2$ | max $\chi^2$ |
|--------------|---------------|--------------|
| **Qwen-2.5-Math (1.5B / 7B)** | 1.48 ± 2.20 | 109.20 |
| **Qwen-3 (1.7B / 14B** | 3.91 ± 12.76 | 155.21 |

While we do not recommend using the bound in Theorem 1 as a practical guidance for choosing hyperparameters, as the theorem is not necessarily tight, it can yield practical values in practice. If, for example, the $\chi^2$ is equal to 2, and we set $\beta = 1$, the bound would guarantee that, if choosing $n \geq (3e^2 - 1)/(e^{0.1} - 1) \approx 201$, the KL between $\pi_{\beta,B}$ and $\tilde{\pi}_{\text{GSI}}$ is bounded by $\epsilon = 0.1$.

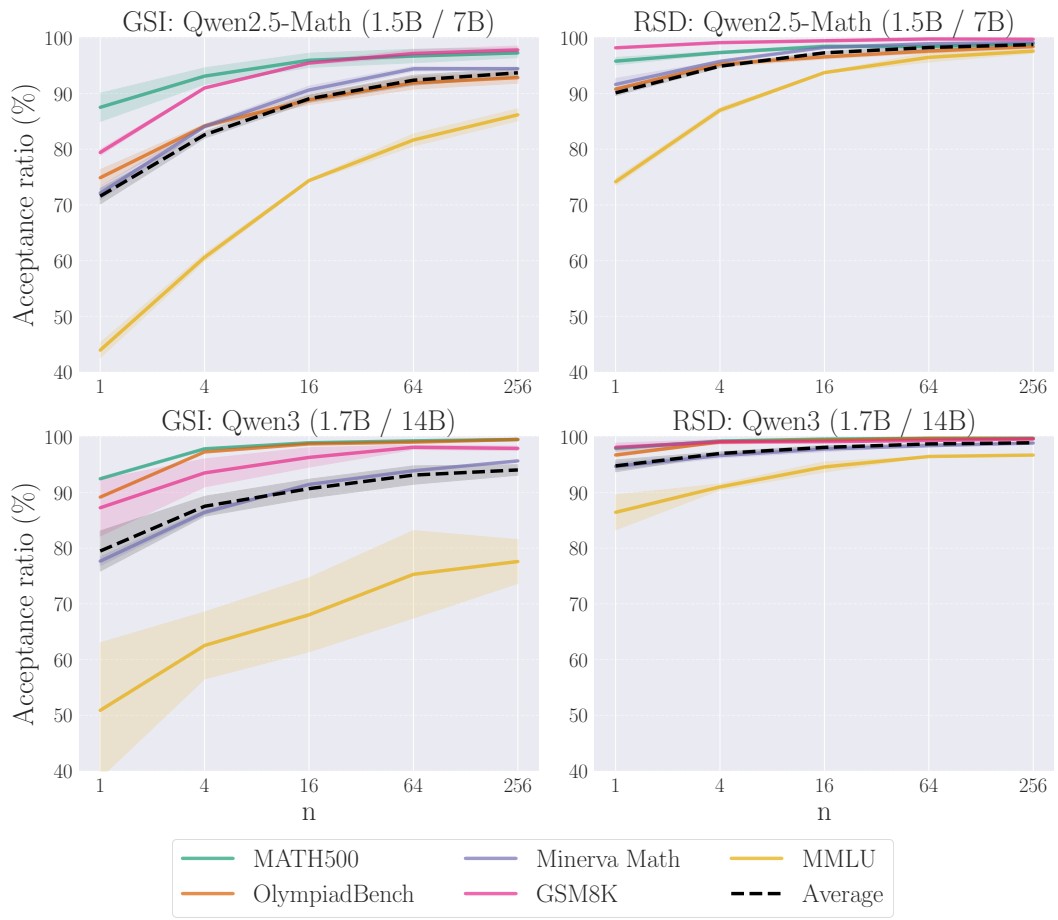

Figure 5: Acceptance ratios for GSI and RSD across datasets and models, with 95% confidence intervals. As $n$ increases, the acceptance ratio of GSI approaches 90%. The acceptance ratio of RSD is much higher and converges to almost 100% as $n$ increases, which means RSD effectively collapses to soft best-of-$n$ with $\pi_S$.

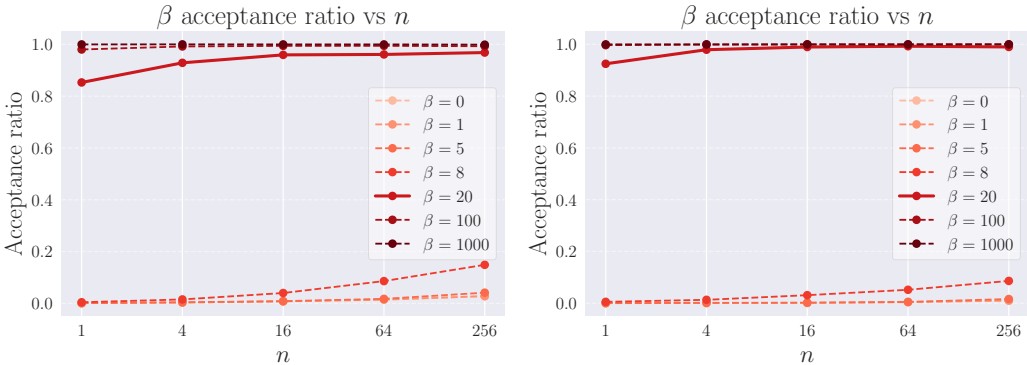

Figure 6: Acceptance ratio of GSI for different values of $\beta$ on MATH500. Left: Qwen2.5-Math; right: Qwen3. A sharp phase transition between $\beta = 8$ and $\beta = 20$ can be observed.

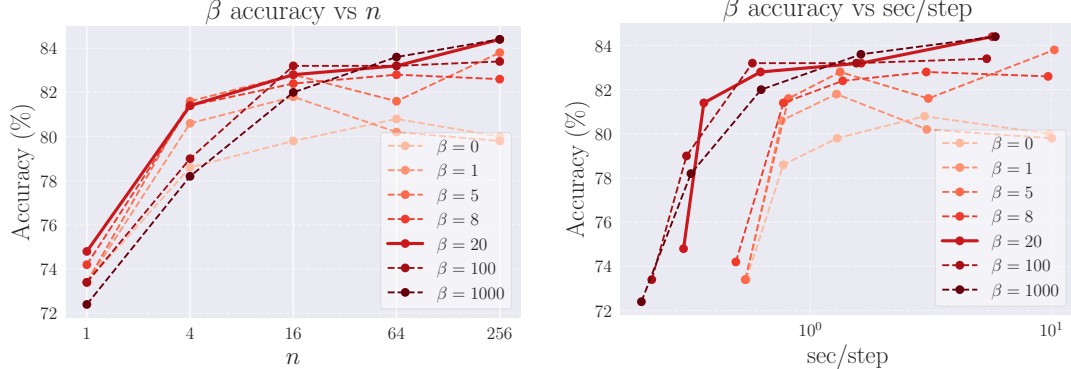

Figure 7: **Qwen2.5-Math:** Accuracy of GSI over $n$ (left) and over seconds per step (right) for different values of $\beta$, on MATH500. In the right plot, each curve corresponds to $n = 1$, 4, 16, 64, 256 for a fixed value of $\beta$ (where each dot on the curve corresponds to one value $n$). Our value $\beta = 20$ performs best overall, but as $n$ varies, different $\beta$ can have an edge. Runtimes reported on H200 GPUs.

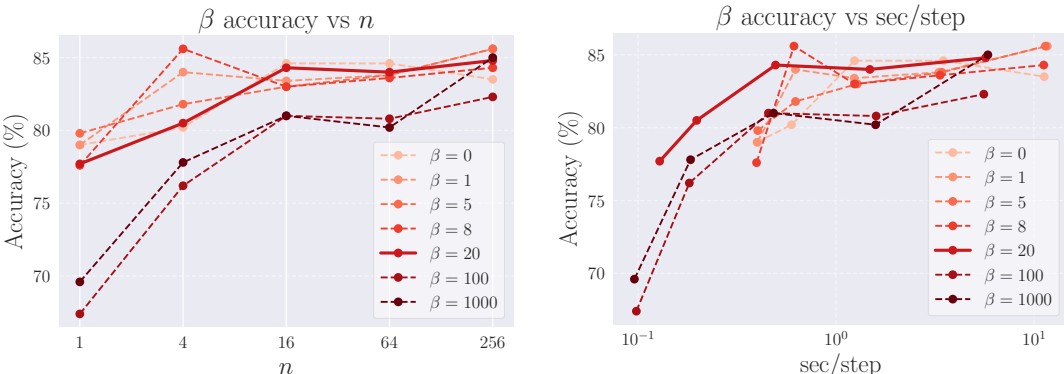

Figure 8: **Qwen3:** Accuracy of GSI over $n$ (left) and over seconds per step (right) for different values of $\beta$, on MATH500. In the right plot, each curve corresponds to $n = 1$, 4, 16, 64, 256 for a fixed value of $\beta$ (where each dot on the curve corresponds to one value $n$). Our value $\beta = 20$ performs best overall, but as $n$ varies, different $\beta$ can have an edge. Runtimes reported on H200 GPUs.

## C.6 RUNTIME COMPARISON

In Tables 5 and 6, we provide extended versions of Table 1 with runtime values across $n$ on H100 GPUs for Qwen2.5-Math, and A100 GPUs for Qwen3.[2]

## C.7 REASONING TRACES

We provide several examples from MATH500 and MMLU-STEM and the reasoning traces generated by GSI and RSD with our Qwen2.5-Math models, in addition to the two examples in the main text. The following boxes contain samples, alongside the reasoning steps selected by the two algorithms (including rejected steps, which are marked by being crossed out) for $n = 4$. For GSI, the last column contains the tilted reward (for samples from $\pi_S$) resp. the normal reward (for samples from $\pi_B$). For RSD it always contains the normal reward. We picked samples where reasoning traces were not too long in order to fit them on one page; note that on average, reasoning traces are much longer (cmp. Table 5).

---

[2]For computational reasons, for the Qwen3 experiements we ran $n = 256$ on H100 and H200 GPUs, hence we do not report them in the table for consistency.

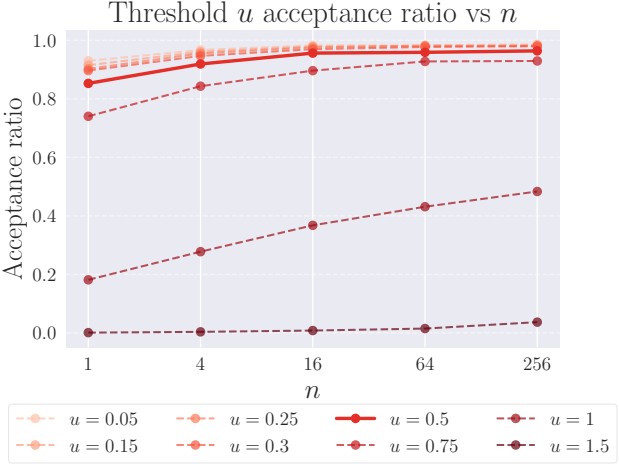

Figure 9: **Larger thresholds $u$ lead to lower acceptance rates in GSI.** We show acceptance ratios of GSI for different acceptance thresholds $u$ on MATH500 for the Qwen2.5-Math models.

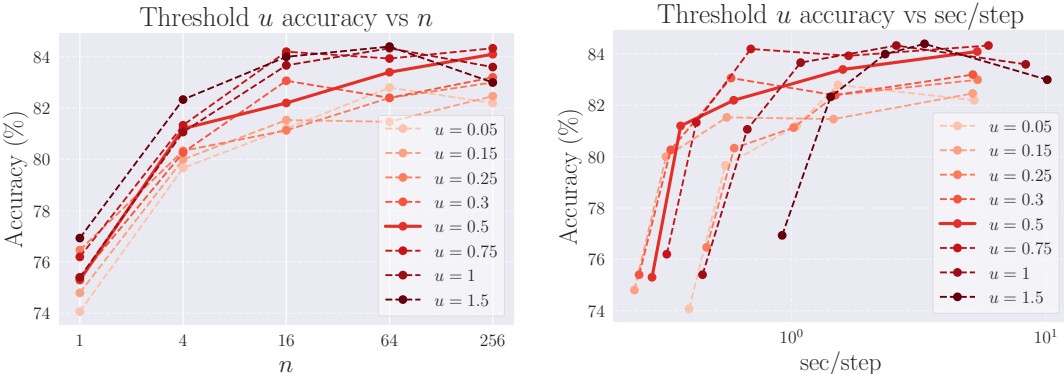

Figure 10: **Left: Larger acceptance thresholds $u$ lead to higher accuracy in GSI.** This is to be expected, as larger thresholds mean higher probability of sampling with $\pi_B$. **Right: When plotting accuracy as a function of seconds per step, no single threshold $u$ performs best.** Each line in this plot corresponds to the values $n = 1, 4, 16, 64, 256$, for a fixed threshold $u$. All plots averaged over MATH500 using the Qwen2.5-Math models.

**MATH500, Example 3.** For this difficult question, GSI repeatedly resamples from the base model to find the right answer. RSD accepts all draft samples and arrives at a wrong answer.

**MATH500, Example 4.** In the fourth example, we see that GSI can sometimes also reject *correct* steps generated by the small model, if the tilted reward is too small. GSI still arrives at the correct answer in the end. In this example, RSD does not produce any final answer.

**MATH500, Example 5.** This example highlights an interesting phenomenon: without any intervention, GSI and RSD generate *almost the exact same reasoning trace*. At a crucial step, $\pi_S$ incorrectly rounds $233/43$ to $5.5$, which GSI corrects by resampling from $\pi_B$ and correctly rounding to four decimals, $5.4186$, while RSD accepts the sample from $\pi_S$ and arrives at a wrong answer. This example also highlights why including the log ratio in the reward can be crucial: The incorrect step under $\pi_S$ receives an (almost) perfect reward of $r = 0.999$ in RSD. The almost identical step in GSI has an (almost) perfect reward of $r = 0.998$ (not depicted in the box), while its tilted reward is only $0.148$.

**MATH500, Example 6.** We show that it can happen that GSI does not solve a problem that RSD manages to solve. However, this only occurred three times in the entire dataset of 500 samples.

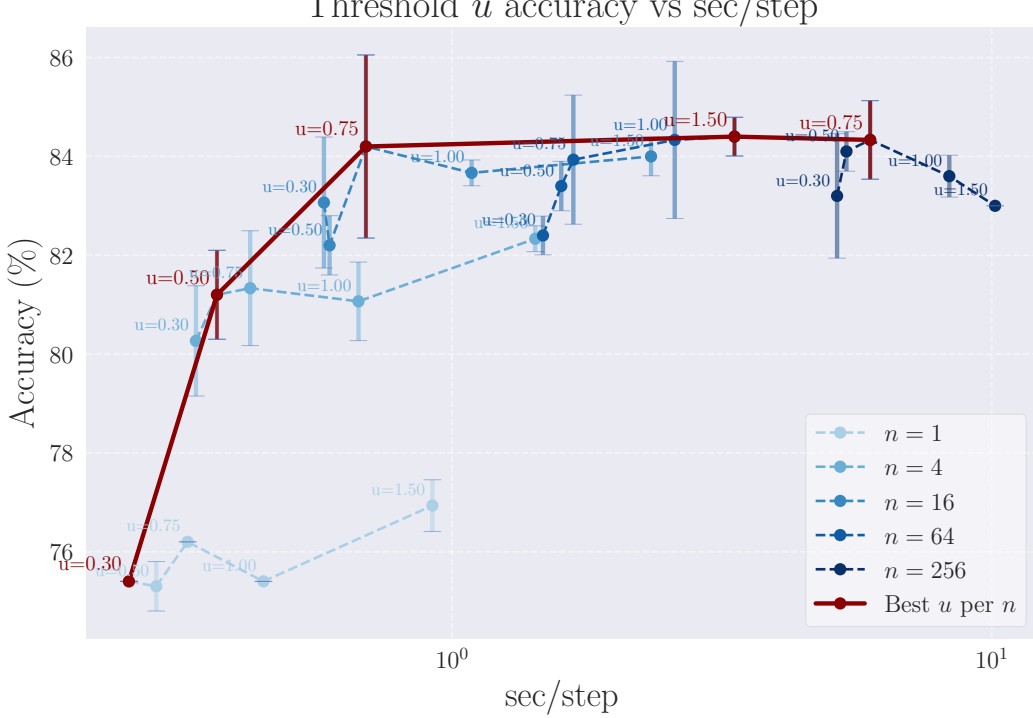

Figure 11: **Empirical Pareto frontier of optimal thresholds** $u$ **for different values of** $n$**. The optimal** $u$ **is a concave function of** $n$. For each value of $n = 1, 4, 16, 64, 256$, we show the average accuracy as a function of seconds per reasoning step for $u = 0.3, 0.5, 0.75, 1.0, 1.5$ with 95% confidence intervals over three random seeds. For the Pareto frontier, we select one value $u$ for each $n$. Averaged over MATH500 using the Qwen2.5-Math models.

**MMLU-STEM, Example 1.** The draft model seems to be generally quite weak on MMLU-STEM and often produces nonsense, including random artifacts such as Chinese and Korean characters. This example shows that GSI can help in mitigating the weaknesses of the small model to some degree. However, several nonsensical steps from the draft model still slip through. Nonetheless, GSI manages to find the correct response, whereas RSD does not.

**MMLU-STEM, Example 2.** As in the previous example, the draft model struggles to produce coherent responses. GSI catches some of its errors, but both GSI and RSD answer this question wrong.

# D  ASSETS

## D.1  HARDWARE

Most of the experiments were run on NVIDIA H100 GPUs. Each model was hosted on its own GPU and implemented with vLLM (Kwon et al., 2023). Some experiments were also run on NVIDIA A100 GPUs and NVIDIA H200 GPUs with the same setup.

## D.2  LIBRARIES

We heavily relied on the following open-source python libraries: PyTorch (Paszke et al., 2019) (license: BSD), transformers by HuggingFace (Wolf et al., 2020) (license: Apache-2.0), and vLLM (Kwon et al., 2023) (license: Apache-2.0).

Table 5: **Qwen2.5 on H100:** Inference time (in seconds) per reasoning step, number of reasoning steps per sample, and percentage of steps accepted (averaged across all datasets, with 95% confidence intervals over three random seeds), for $n = 1, 4, 16, 64, 256$ (extension of Table 1).

| n | Method | s / step ($\downarrow$) | # steps | % accept | steps / s ($\uparrow$) |
|---|---|---|---|---|---|
| 1 | GSI (ours) | $0.33 \pm 0.02$ | $8.9 \pm 0.1$ | $65.4 \pm 0.1$ | $3.03 \pm 0.17$ |
| | RSD | $0.24 \pm 0.01$ | $8.8 \pm 0.2$ | $90.1 \pm 0.0$ | $4.17 \pm 0.17$ |
| | S-BoN (small) | $0.20 \pm 0.00$ | $8.7 \pm 0.1$ | – | $5.00 \pm 0.00$ |
| | S-BoN (base) | $0.39 \pm 0.03$ | $9.3 \pm 0.2$ | – | $2.56 \pm 0.18$ |
| 4 | GSI (ours) | $0.43 \pm 0.03$ | $10.6 \pm 0.3$ | $76.7 \pm 0.1$ | $2.33 \pm 0.15$ |
| | RSD | $0.34 \pm 0.01$ | $9.7 \pm 0.1$ | $94.9 \pm 0.0$ | $2.94 \pm 0.08$ |
| | S-BoN (small) | $0.32 \pm 0.01$ | $9.6 \pm 0.0$ | – | $3.12 \pm 0.09$ |
| | S-BoN (base) | $0.57 \pm 0.01$ | $10.2 \pm 0.3$ | – | $1.75 \pm 0.03$ |
| 16 | GSI (ours) | $0.72 \pm 0.05$ | $11.4 \pm 0.2$ | $82.0 \pm 0.1$ | $1.39 \pm 0.09$ |
| | RSD | $0.61 \pm 0.01$ | $10.3 \pm 0.3$ | $97.3 \pm 0.0$ | $1.64 \pm 0.03$ |
| | S-BoN (small) | $0.52 \pm 0.03$ | $10.3 \pm 0.1$ | – | $1.92 \pm 0.10$ |
| | S-BoN (base) | $0.94 \pm 0.03$ | $10.5 \pm 0.2$ | – | $1.06 \pm 0.03$ |
| 64 | GSI (ours) | $1.78 \pm 0.12$ | $12.0 \pm 0.4$ | $84.3 \pm 0.1$ | $0.56 \pm 0.04$ |
| | RSD | $1.60 \pm 0.03$ | $10.9 \pm 0.3$ | $98.2 \pm 0.1$ | $0.62 \pm 0.01$ |
| | S-BoN (small) | $1.50 \pm 0.04$ | $10.7 \pm 0.1$ | – | $0.67 \pm 0.01$ |
| | S-BoN (base) | $1.99 \pm 0.07$ | $10.8 \pm 0.2$ | – | $0.50 \pm 0.02$ |
| 256 | GSI (ours) | $5.80 \pm 0.23$ | $13.0 \pm 0.3$ | $93.6 \pm 0.0$ | $0.17 \pm 0.01$ |
| | RSD | $5.52 \pm 0.42$ | $11.3 \pm 1.0$ | $99.1 \pm 0.0$ | $0.18 \pm 0.01$ |
| | S-BoN (small) | $5.46 \pm 0.10$ | $11.3 \pm 0.1$ | – | $0.18 \pm 0.00$ |
| | S-BoN (base) | $5.88 \pm 0.11$ | $11.1 \pm 0.3$ | – | $0.17 \pm 0.00$ |

### D.3 CODE REPOSITORY

We used the RewardHub library by Red Hat AI Innovation Team, and grading functions from OpenAI's prm800k repository to extract and grade answers from LLM-generated responses.

## E USE OF LARGE LANGUAGE MODELS

We utilized generative AI tools for code generation and debugging. The authors carried out all of the substantive research contributions, experiments, and proofs.

Table 6: **Qwen3 on A100:** Inference time (in seconds) per reasoning step, number of reasoning steps per sample, and percentage of steps accepted (averaged across all datasets, with $95\%$ confidence intervals over three random seeds), for $n = 1, 4, 16, 64$ (extension of Table 1).

| n | Method | s / step ($\downarrow$) | # steps | % accept | steps / s ($\uparrow$) |
|---|---|---|---|---|---|
| 1 | GSI (ours) | $0.35 \pm 0.02$ | $24.1 \pm 0.0$ | $80.9 \pm 0.1$ | $2.85 \pm 0.15$ |
| | RSD | $0.24 \pm 0.01$ | $25.2 \pm 0.1$ | $95.3 \pm 0.1$ | $4.17 \pm 0.17$ |
| | S-BoN (s) | $0.2 \pm 0.00$ | $23.3$ | $-$ | $5.00 \pm 0.04$ |
| | S-BoN (b) | $0.59 \pm 0.01$ | $23.3$ | $-$ | $1.69 \pm 0.03$ |
| 4 | GSI (ours) | $0.56 \pm 0.04$ | $26.7 \pm 0.3$ | $88.0 \pm 0.1$ | $1.79 \pm 0.12$ |
| | RSD | $0.4 \pm 0.01$ | $28.1 \pm 0.1$ | $97.2 \pm 0.1$ | $2.50 \pm 0.06$ |
| | S-BoN (s) | $0.38 \pm 0.01$ | $24.8 \pm 0.2$ | $-$ | $2.63 \pm 0.07$ |
| | S-BoN (b) | $0.83 \pm 0.02$ | $24.9 \pm 0.1$ | $-$ | $1.20 \pm 0.03$ |
| 16 | GSI (ours) | $1.21 \pm 0.05$ | $27.5 \pm 0.4$ | $91.5 \pm 0.1$ | $0.83 \pm 0.03$ |
| | RSD | $0.96 \pm 0.01$ | $29.0 \pm 0.2$ | $98.2 \pm 0.1$ | $1.04 \pm 0.01$ |
| | S-BoN (s) | $1.01 \pm 0.00$ | $26.1 \pm 0.2$ | $-$ | $0.99 \pm 0.00$ |
| | S-BoN (b) | $1.82 \pm 0.06$ | $24.8 \pm 0.3$ | $-$ | $0.55 \pm 0.02$ |
| 64 | GSI (ours) | $3.26 \pm 0.11$ | $28.8 \pm 0.8$ | $93.1 \pm 0.0$ | $0.31 \pm 0.01$ |
| | RSD | $2.93 \pm 0.05$ | $29.2 \pm 0.7$ | $98.7 \pm 0.2$ | $0.34 \pm 0.01$ |
| | S-BoN (s) | $3.01 \pm 0.05$ | $26.4 \pm 1.1$ | $-$ | $0.33 \pm 0.01$ |
| | S-BoN (b) | $4.75 \pm 0.06$ | $25.2 \pm 0.3$ | $-$ | $0.21 \pm 0.00$ |

---

**GSI:** MATH500, Example 3 (Qwen2.5-Math)

---

**Prompt:** The wheel shown is spun twice, so that the numbers indicated by the pointer are randomly determined (with each number on the wheel being equally likely). The two numbers determined in this way are recorded. The first number is divided by 4, determining one of the remainders 1,2,3 marking the columns of the checkerboard shown. The second number is divided by 5, determining one of the remainders 1,2,3,4 marking the rows of the checkerboard. Finally, a checker is placed on the square where this column and row meet. What is the probability that the checker is placed on a shaded square of the checkerboard?
[asy]  unitsize(1cm);  draw(Circle((0,0),2),linewidth(0.7));  draw((1.7,1)--(-1.7,-1),linewidth(0.7));  draw((1.7,-1)--(-1.7,1),linewidth(0.7));  draw((0,2)--(0,-2));  label("1",(0.8,0.5),NW);  label("2",(0.8,-0.5),SW);  label("6",(-0.8,0.5),NE); label("9",(-0.8,-0.5),SE);  label("3",(-0.7,0),W);  label("7",(0.7,0),E);  draw((-2.8,0)--(-2.1,0),Arrow);  label("Pointer",(-2.8,0),W);  fill((3,0)--(3,1)--(4,1)--(4,0)--cycle,gray(0.7));  fill((3,-2)--(3,-1)--(4,-1)--(4,-2)--cycle,gray(0.7)); fill((4,1)--(4,2)--(5,2)--(5,1)--cycle,gray(0.7));  fill((4,-1)--(4,0)--(5,0)--(5,-1)--cycle,gray(0.7));  fill((5,0)--(5,1)--(6,1)--(6,0)--cycle,gray(0.7)); fill((5,-2)--(5,-1)--(6,-1)--(6,-2)--cycle,gray(0.7));  draw((3,-2)--(3,2)--(6,2)--(6,-2)--cycle,linewidth(0.7));  draw((3,-1)--(6,-1),linewidth(0.7));  draw((3,0)--(6,0),linewidth(0.7));  draw((3,1)--(6,1),linewidth(0.7));  draw((4,-2)--(4,2),linewidth(0.7)); draw((5,-2)--(5,2),linewidth(0.7));  label("1",(3.5,-2),S);  label("2",(4.5,-2),S);  label("3",(5.5,-2),S);  label("1",(3,-1.5),W); label("2",(3,-0.5),W); label("3",(3,0.5),W); label("4",(3,1.5),W); [/asy]

| Step | Generated Reasoning Steps ( **Draft** $\pi_S$ , **Target** $\pi_B$ ) | $\tilde{r}(y_{i*}^S)/r(y_{j*}^B)$ |
|---|---|---|
| 1 | To determine the probability that the checker is placed on a shaded square of the checkerboard, we need to analyze the outcomes of the two spins and the resulting positions on the checkerboard. | 0.940 |
| 2 | First, let's identify the possible outcomes for each spin: - The first spin can result in the numbers 1, 2, 3, 4, 5, or 6. - The second spin can result in the numbers 1, 2, 3, 4, 5. | 0.047 |
|  | First, let's identify the possible remainders when the numbers on the wheel are divided by 4 and 5: | 0.990 |
| 3 | - For division by 4, the possible remainders are 1, 2, 3, and 4. - For division by 5, the possible remainders are 1, 2, 3, 4, and 5. | 0.542 |
| 4 | However, the problem only asks about the remainders 1, 2, and 3 for the first spin (divided by 4) and 1, 2, 3, and 4 for the second spin (divided by 5). This means we are only considering the first three remainders for the first spin and all four remainders for the second spin. | -0.706 |
|  | The checkerboard has a total of $4 \times 5 = 20$ squares, arranged in 4 columns and 5 rows. We need to count how many of these squares are shaded. | 0.980 |
| 5 | From the diagram, we observe that the shaded squares are: - (1,1), (1,3), (1,4), - (2,2), (2,4), - (3,1), (3,3), - (4,2), (4,4). | -0.128 |
|  | From the checkerboard, we observe that the shaded squares follow a pattern where the squares in the first and third columns are shaded in the first and third rows, and the squares in the second and fourth columns are shaded in the second and fourth rows. This gives us the following shaded squares: | 0.803 |
| 6 | Shaded   Unshaded   Shaded   Unshaded   Shaded
Unshaded   Shaded   Unshaded   Shaded   Unshaded
Shaded   Unshaded   Shaded   Unshaded   Shaded
Unshaded   Shaded   Unshaded   Shaded   Unshaded | 0.543 |
| 7 | Counting the shaded squares, we find there are 10 shaded squares out of the 20 squares on the checkerboard. | 1.132 |
| 8 | Since each number on the wheel is equally likely, each remainder when divided by 4 and 5 is equally likely. Therefore, each of the 20 possible outcomes is equally likely. | 0.402 |
|  | The probability that the checker is placed on a shaded square is the ratio of the number of shaded squares to the total number of squares: | 0.975 |
| 9 | $$\text{Probability} = \frac{\text{Number of shaded squares}}{\text{Total number of squares}} = \frac{10}{20} = \frac{1}{2}$$ | 1.000 |
| 10 | Thus, the probability that the checker is placed on a shaded square of the checkerboard is $\boxed{\dfrac{1}{2}}$. | 1.053 |

**Correct Answer found**: True

**RSD:** MATH500, Example 3 (Qwen2.5-Math)

**Prompt:** The wheel shown is spun twice, so that the numbers indicated by the pointer are randomly determined (with each number on the wheel being equally likely). The two numbers determined in this way are recorded. The first number is divided by 4, determining one of the remainders 1,2,3 marking the columns of the checkerboard shown. The second number is divided by 5, determining one of the remainders 1,2,3,4 marking the rows of the checkerboard. Finally, a checker is placed on the square where this column and row meet. What is the probability that the checker is placed on a shaded square of the checkerboard? [asy] unitsize(1cm); draw(Circle((0,0),2),linewidth(0.7)); draw((1.7,1)–(-1.7,-1),linewidth(0.7)); draw((1.7,-1)–(-1.7,1),linewidth(0.7)); draw((0,2)–(0,-2)); label("1",(0.8,0.5),NW); label("2",(0.8,-0.5),SW); label("6",(-0.8,0.5),NE); label("9",(-0.8,-0.5),SE); label("3",(-0.7,0),W); label("7",(0.7,0),E); draw((-2.8,0)–(-2.1,0),Arrow); label("Pointer",(-2.8,0),W); fill((3,0)–(3,1)–(4,1)–(4,0)–cycle,gray(0.7)); fill((3,-2)–(3,-1)–(4,-1)–(4,-2)–cycle,gray(0.7)); fill((4,1)–(4,2)–(5,2)–(5,1)–cycle,gray(0.7)); fill((4,-1)–(4,0)–(5,0)–(5,-1)–cycle,gray(0.7)); fill((5,0)–(5,1)–(6,1)–(6,0)–cycle,gray(0.7)); fill((5,-2)–(5,-1)–(6,-1)–(6,-2)–cycle,gray(0.7)); draw((3,-2)–(3,2)–(6,2)–(6,-2)–cycle,linewidth(0.7)); draw((3,-1)–(6,-1),linewidth(0.7)); draw((3,0)–(6,0),linewidth(0.7)); draw((3,1)–(6,1),linewidth(0.7)); draw((4,-2)–(4,2),linewidth(0.7)); draw((5,-2)–(5,2),linewidth(0.7)); label("1",(3.5,-2),S); label("2",(4.5,-2),S); label("3",(5.5,-2),S); label("1",(3,-1.5),W); label("2",(3,-0.5),W); label("3",(3,0.5),W); label("4",(3,1.5),W); [/asy]

| Step | Generated Reasoning Steps ( **Draft** $\pi_S$ , **Target** $\pi_B$ ) | $r(y_{i*}^S)/r(y_{j*}^B)$ |
|------|------|------|
| 1 | To determine the probability that the checker is placed on a shaded square of the checkerboard, we need to analyze the structure of the checkerboard and the possible outcomes of the spin. | 0.996 |
| 2 | First, let's identify the shaded squares on the checkerboard. The checkerboard is a 3x4 grid, with the following shaded squares: - (1,1) - (1,3) - (2,2) - (2,4) | 0.784 |
| 3 | This means there are 4 shaded squares out of a total of 12 squares. | 0.967 |
| 4 | Next, we need to determine the probability of landing on a shaded square. The wheel is spun twice, so there are 6 possible outcomes for each spin (1, 2, 3, 4, 5, 6). When the first number is divided by 4, the possible remainders are 1, 2, 3. Similarly, when the second number is divided by 5, the possible remainders are 1, 2, 3, 4. Therefore, there are $3 \times 4 = 12$ possible outcomes in total. | 0.895 |
| 5 | Now, we need to count the number of outcomes that result in a shaded square. From the checkerboard, we can see that: - The first spin landing on 1 (remainder 1) and the second spin landing on 1 (remainder 1) results in the shaded square (1,1). - The first spin landing on 1 (remainder 1) and the second spin landing on 3 (remainder 3) results in the shaded square (1,3). - The first spin landing on 2 (remainder 2) and the second spin landing on 2 (remainder 2) results in the shaded square (2,2). - The first spin landing on 2 (remainder 2) and the second spin landing on 4 (remainder 4) results in the shaded square (2,4). | 0.947 |
| 6 | So, there are 4 outcomes that result in a shaded square. The probability is the number of favorable outcomes divided by the total number of outcomes, which is $\frac{4}{12} = \frac{1}{3}$. | 0.967 |
| 7 | Therefore, the probability that the checker is placed on a shaded square is $\boxed{\dfrac{1}{3}}$. | 1.000 |

**Correct Answer found**: False

**GSI:** MATH500, Example 4 (Qwen2.5-Math)

**Prompt:** Let $z$ be a complex number such that
$$z + \frac{1}{z} = \frac{1 + \sqrt{5}}{2}.$$
Find
$$z^{85} + \frac{1}{z^{85}}.$$

| Step | Generated Reasoning Steps ( **Draft** $\pi_S$ , **Target** $\pi_B$ ) | $\tilde{r}(y_{i*}^S)/r(y_{j*}^B)$ |
|---|---|---|
| 1 | Given the equation $z + \frac{1}{z} = \frac{1+\sqrt{5}}{2}$, we recognize that $z$ is a complex number on the unit circle, meaning $|z| = 1$. Therefore, we can write $z = e^{i\theta}$ for some real number $\theta$. | 0.530 |
| 2 | First, we express $z + \frac{1}{z}$ in terms of $\theta$: $$z + \frac{1}{z} = e^{i\theta} + e^{-i\theta} = 2\cos\theta.$$ Given that $z + \frac{1}{z} = \frac{1+\sqrt{5}}{2}$, we have: $$2\cos\theta = \frac{1 + \sqrt{5}}{2}.$$ Solving for $\cos\theta$, we get: $$\cos\theta = \frac{1 + \sqrt{5}}{4}.$$ | 0.412 |
| | The given equation can be rewritten as: $$z + \frac{1}{z} = e^{i\theta} + e^{-i\theta} = 2\cos\theta.$$ We know that $2\cos\theta = \frac{1+\sqrt{5}}{2}$, so: $$\cos\theta = \frac{1 + \sqrt{5}}{4}.$$ The angle $\theta$ that satisfies this equation is $\theta = \pm\frac{\pi}{5}$. Therefore, $z$ can be either $e^{i\pi/5}$ or $e^{-i\pi/5}$. | 0.972 |
| 3 | We need to find $z^{85} + \frac{1}{z^{85}}$. Let's consider the two cases separately. | 0.830 |
| 4 | 1. If $z = e^{i\pi/5}$, then: $$z^{85} = \left(e^{i\pi/5}\right)^{85} = e^{i85\pi/5} = e^{i17\pi} = e^{i(16\pi+\pi)} = e^{i\pi} = -1.$$ So, $$z^{85} + \frac{1}{z^{85}} = -1 + \frac{1}{-1} = -1 - 1 = -2.$$ | 0.942 |
| 5 | 2. If $z = e^{-i\pi/5}$, then: $$z^{85} = \left(e^{-i\pi/5}\right)^{85} = e^{-i85\pi/5} = e^{-i17\pi} = e^{-i(16\pi+\pi)} = e^{-i\pi} = -1.$$ So, $$z^{85} + \frac{1}{z^{85}} = -1 + \frac{1}{-1} = -1 - 1 = -2.$$ | 0.998 |
| 6 | In both cases, we get the same result. Therefore, the value of $z^{85} + \frac{1}{z^{85}}$ is: $$\boxed{-2}.$$ | 0.998 |

**Correct Answer found**: True

**RSD:** MATH500, Example 4 (Qwen2.5-Math)

**Prompt:** Let $z$ be a complex number such that
$$z + \frac{1}{z} = \frac{1 + \sqrt{5}}{2}.$$
Find
$$z^{85} + \frac{1}{z^{85}}.$$

| Step | Generated Reasoning Steps ( Draft $\pi_S$ , Target $\pi_B$ ) | $r(y_{i*}^S)/r(y_{j*}^B)$ |
|---|---|---|
| 1 | To find $z^{85} + \frac{1}{z^{85}}$ given the equation $z + \frac{1}{z} = \frac{1+\sqrt{5}}{2}$, we start by recognizing that $z + \frac{1}{z}$ is the form of the golden ratio $\phi = \frac{1+\sqrt{5}}{2}$. | 0.979 |
| 2 | First, let's solve for $z$. Let $z = e^{i\theta}$, where $\theta$ is a real number. Then, $$z + \frac{1}{z} = e^{i\theta} + e^{-i\theta} = 2\cos\theta.$$ Given that $z + \frac{1}{z} = \frac{1+\sqrt{5}}{2}$, we have: $$2\cos\theta = \frac{1+\sqrt{5}}{2}.$$ Thus, $$\cos\theta = \frac{1+\sqrt{5}}{4}.$$ | 0.970 |
| 3 | Next, we need to find $z^{85} + \frac{1}{z^{85}}$. Using the exponential form, we have: $$z^{85} = e^{i85\theta} \quad \text{and} \quad \frac{1}{z^{85}} = e^{-i85\theta}.$$ Therefore, $$z^{85} + \frac{1}{z^{85}} = e^{i85\theta} + e^{-i85\theta} = 2\cos(85\theta).$$ | 0.993 |
| 4 | To find $\cos(85\theta)$, we use the angle addition formula for cosine: $$\cos(85\theta) = \cos(84\theta + \theta) = \cos(84\theta)\cos(\theta) - \sin(84\theta)\sin(\theta).$$ | 0.977 |
| 5 | We need to determine $\cos(84\theta)$ and $\sin(84\theta)$. Notice that: $$\cos(2\theta) = 2\cos^2(\theta) - 1.$$ Substituting $\cos\theta = \frac{1+\sqrt{5}}{4}$, we get: $$\cos^2(\theta) = \left(\frac{1+\sqrt{5}}{4}\right)^2 = \frac{(1+\sqrt{5})^2}{16} = \frac{1+2\sqrt{5}+5}{16} = \frac{6+2\sqrt{5}}{16} = \frac{3+\sqrt{5}}{8}.$$ Thus, $$\cos(2\theta) = 2 \cdot \frac{3+\sqrt{5}}{8} - 1 = \frac{6+2\sqrt{5}}{8} - 1 = \frac{6+2\sqrt{5}-8}{8} = \frac{-2+2\sqrt{5}}{8} = \frac{-1+\sqrt{5}}{4}.$$ | 0.959 |
| 6 | Using similar steps, we can find $\cos(4\theta)$, $\cos(8\theta)$, and so on, but this becomes increasingly complex. Instead, we recognize a pattern in the powers of $\cos$ and $\sin$ that simplifies our work. We use the fact that $\cos(2\theta)$ is related to $\cos(\theta)$ and $\sin(\theta)$, and we can use the properties of the golden ratio. | 0.821 |

**Correct Answer found**: False

**GSI:** MATH500, Example 5 (Qwen2.5-Math)

**Prompt: Juan's Old Stamping Grounds**
Juan organizes the stamps in his collection by country and by the decade in which they were issued. The prices he paid for them at a stamp shop were: Brazil and France, 6 cents each, Peru 4 cents each, and Spain 5 cents each. (Brazil and Peru are South American countries and France and Spain are in Europe.) [asy] /* AMC8 2002 #8, 9, 10 Problem */ size(3inch, 1.5inch); for ( int y = 0; y ¡= 5; ++y )
draw((0,y)–(18,y));      draw((0,0)–(0,5));  draw((6,0)–(6,5));  draw((9,0)–(9,5));  draw((12,0)–(12,5));  draw((15,0)–(15,5));  draw((18,0)–(18,5));
label(scale(0.8)*"50s", (7.5,4.5));  label(scale(0.8)*"4", (7.5,3.5));  label(scale(0.8)*"8", (7.5,2.5));  label(scale(0.8)*"6", (7.5,1.5)); label(scale(0.8)*"3", (7.5,0.5));
label(scale(0.8)*"60s", (10.5,4.5));  label(scale(0.8)*"7", (10.5,3.5));  label(scale(0.8)*"4", (10.5,2.5));  label(scale(0.8)*"4", (10.5,1.5)); label(scale(0.8)*"9", (10.5,0.5));
label(scale(0.8)*"70s", (13.5,4.5));  label(scale(0.8)*"12", (13.5,3.5));  label(scale(0.8)*"12", (13.5,2.5));  label(scale(0.8)*"6", (13.5,1.5)); label(scale(0.8)*"13", (13.5,0.5));
label(scale(0.8)*"80s", (16.5,4.5));  label(scale(0.8)*"8", (16.5,3.5));  label(scale(0.8)*"15", (16.5,2.5));  label(scale(0.8)*"10", (16.5,1.5)); label(scale(0.8)*"9", (16.5,0.5));
label(scale(0.8)*"Country", (3,4.5));  label(scale(0.8)*"Brazil", (3,3.5));  label(scale(0.8)*"France", (3,2.5)); label(scale(0.8)*"Peru", (3,1.5)); label(scale(0.8)*"Spain", (3,0.5));
label(scale(0.9)*"Juan's Stamp Collection", (9,0), S); label(scale(0.9)*"Number of Stamps by Decade", (9,5), N); [/asy] What was the average price, in cents, of his 70's stamps? Round your answer to the nearest tenth of a cent.

| Step | Generated Reasoning Steps ( **Draft** $\pi_S$ , **Target** $\pi_B$ ) | $\tilde{r}(y_{i*}^S)/r(y_{j*}^B)$ |
|---|---|---|
| 1 | To find the average price of Juan's 70's stamps, we need to follow these steps: | 0.899 |
| 2 | 1. **Identify the number of stamps from each country in the 70's:** - Brazil: 12 stamps - France: 12 stamps - Peru: 6 stamps - Spain: 13 stamps | 0.892 |
| 3 | 2. **Calculate the total number of stamps in the 70's:** $$12 + 12 + 6 + 13 = 43 \text{ stamps}$$ | 0.636 |
| 4 | 3. **Determine the total cost of the stamps from each country:** - Brazil: 12 stamps $\times$ 6 cents per stamp $=$ 72 cents - France: 12 stamps $\times$ 6 cents per stamp $=$ 72 cents - Peru: 6 stamps $\times$ 4 cents per stamp $=$ 24 cents - Spain: 13 stamps $\times$ 5 cents per stamp $=$ 65 cents | 0.859 |
| 5 | 4. **Calculate the total cost of all the 70's stamps:** $$72 + 72 + 24 + 65 = 233 \text{ cents}$$ | 1.032 |
| 6 | 5. **Find the average price per stamp:** $$\text{Average price} = \frac{\text{Total cost}}{\text{Total number of stamps}} = \frac{233 \text{ cents}}{43 \text{ stamps}}$$ | 1.039 |
| 7 | 6. **Perform the division and round to the nearest tenth of a cent:** $$\frac{233}{43} \approx 5.488$$ Rounding 5.488 to the nearest tenth gives us 5.5. | 0.148 |
| | 6. **Perform the division:** $$\frac{233}{43} \approx 5.4186$$ | 0.996 |
| 8 | 7. **Round the result to the nearest tenth of a cent:** $$5.4186 \approx 5.4$$ | 1.027 |
| 9 | Therefore, the average price of Juan's 70's stamps is $\boxed{5.4}$ cents. | 1.013 |

**Correct Answer found**: True

**RSD:** MATH500, Example 5 (Qwen2.5-Math)

**Prompt: Juan's Old Stamping Grounds**
Juan organizes the stamps in his collection by country and by the decade in which they were issued. The prices he paid for them at a stamp shop were: Brazil and France, 6 cents each, Peru 4 cents each, and Spain 5 cents each. (Brazil and Peru are South American countries and France and Spain are in Europe.) [asy] /* AMC8 2002 #8, 9, 10 Problem */ size(3inch, 1.5inch); for ( int y = 0; y ¡= 5; ++y )
draw((0,y)–(18,y));    draw((0,0)–(0,5)); draw((6,0)–(6,5)); draw((9,0)–(9,5)); draw((12,0)–(12,5)); draw((15,0)–(15,5)); draw((18,0)–(18,5));
label(scale(0.8)*"50s", (7.5,4.5)); label(scale(0.8)*"4", (7.5,3.5)); label(scale(0.8)*"8", (7.5,2.5)); label(scale(0.8)*"6", (7.5,1.5)); label(scale(0.8)*"3", (7.5,0.5));
label(scale(0.8)*"60s", (10.5,4.5)); label(scale(0.8)*"7", (10.5,3.5)); label(scale(0.8)*"4", (10.5,2.5)); label(scale(0.8)*"4", (10.5,1.5)); label(scale(0.8)*"9", (10.5,0.5));
label(scale(0.8)*"70s", (13.5,4.5)); label(scale(0.8)*"12", (13.5,3.5)); label(scale(0.8)*"12", (13.5,2.5)); label(scale(0.8)*"6", (13.5,1.5)); label(scale(0.8)*"13", (13.5,0.5));
label(scale(0.8)*"80s", (16.5,4.5)); label(scale(0.8)*"8", (16.5,3.5)); label(scale(0.8)*"15", (16.5,2.5)); label(scale(0.8)*"10", (16.5,1.5)); label(scale(0.8)*"9", (16.5,0.5));
label(scale(0.8)*"Country", (3,4.5));    label(scale(0.8)*"Brazil", (3,3.5));    label(scale(0.8)*"France", (3,2.5));    label(scale(0.8)*"Peru", (3,1.5)); label(scale(0.8)*"Spain", (3,0.5));
label(scale(0.9)*"Juan's Stamp Collection", (9,0), S); label(scale(0.9)*"Number of Stamps by Decade", (9,5), N); [/asy] What was the average price, in cents, of his 70's stamps? Round your answer to the nearest tenth of a cent.

| Step | Generated Reasoning Steps ( **Draft** $\pi_S$ , **Target** $\pi_B$ ) | $r(y_{i*}^S)/r(y_{j*}^B)$ |
|---|---|---|
| 1 | To determine the average price of Juan's 70s stamps, we need to follow these steps: | 0.986 |
| 2 | 1. **Identify the number of stamps from each country in the 70s:** - Brazil: 12 stamps - France: 12 stamps - Peru: 6 stamps - Spain: 13 stamps | 0.993 |
| 3 | 2. **Calculate the total number of 70s stamps:** $$12 + 12 + 6 + 13 = 43 \text{ stamps}$$ | 0.999 |
| 4 | 3. **Determine the total cost of the stamps from each country:** - Brazil: 12 stamps $\times$ 6 cents/stamp $= 72$ cents - France: 12 stamps $\times$ 6 cents/stamp $= 72$ cents - Peru: 6 stamps $\times$ 4 cents/stamp $= 24$ cents - Spain: 13 stamps $\times$ 5 cents/stamp $= 65$ cents | 0.999 |
| 5 | 4. **Calculate the total cost of all the 70s stamps:** $$72 + 72 + 24 + 65 = 233 \text{ cents}$$ | 0.999 |
| 6 | 5. **Find the average price per stamp:** $$\text{Average price} = \frac{\text{Total cost}}{\text{Total number of stamps}} = \frac{233 \text{ cents}}{43 \text{ stamps}}$$ | 1.000 |
| 7 | 6. **Perform the division to find the average price:** $$\frac{233}{43} \approx 5.488372$$ | 0.999 |
| 8 | 7. **Round the result to the nearest tenth of a cent:** $$5.488372 \approx 5.5 \text{ cents}$$ | 0.999 |
| 9 | Thus, the average price of Juan's 70s stamps is $\boxed{5.5}$ cents. | 1.000 |

**Correct Answer found**: False

---

**GSI:** MATH500, Example 6 (Qwen2.5-Math)

**Prompt:** Evaluate
$$\sin(\arcsin 0.4 + \arcsin 0.5) \cdot \sin(\arcsin 0.5 - \arcsin 0.4).$$

| Step | Generated Reasoning Steps ( **Draft** $\pi_S$ , **Target** $\pi_B$ ) | $\tilde{r}(y_{i*}^S)/r(y_{j*}^B)$ |
|---|---|---|
| 1 | To evaluate $\sin(\arcsin 0.4 + \arcsin 0.5) \cdot \sin(\arcsin 0.5 - \arcsin 0.4)$, we start by letting $\alpha = \arcsin 0.4$ and $\beta = \arcsin 0.5$. This means that $\sin \alpha = 0.4$ and $\sin \beta = 0.5$. We need to find $\sin(\alpha+\beta) \cdot \sin(\beta-\alpha)$. | 1.033 |
| 2 | First, we use the angle addition formulas for sine: $$\sin(\alpha + \beta) = \sin \alpha \cos \beta + \cos \alpha \sin \beta$$ $$\sin(\beta - \alpha) = \sin \beta \cos \alpha - \cos \beta \sin \alpha$$ | 0.231 |
| | Using the sine addition and subtraction formulas, we have: $$\sin(\alpha + \beta) = \sin \alpha \cos \beta + \cos \alpha \sin \beta$$ $$\sin(\beta - \alpha) = \sin \beta \cos \alpha - \cos \alpha \sin \beta$$ Multiplying these two expressions together, we get: $$\sin(\alpha + \beta) \cdot \sin(\beta - \alpha) = (\sin \alpha \cos \beta + \cos \alpha \sin \beta)(\sin \beta \cos \alpha - \cos \alpha \sin \beta)$$ This is a difference of squares, so it simplifies to: $$\sin(\alpha + \beta) \cdot \sin(\beta - \alpha) = (\sin \alpha \cos \beta)^2 - (\cos \alpha \sin \beta)^2$$ We know that $\cos^2 \theta = 1 - \sin^2 \theta$ for any angle $\theta$. Therefore, we can substitute $\cos^2 \alpha = 1 - \sin^2 \alpha$ and $\cos^2 \beta = 1 - \sin^2 \beta$ into the equation: $$\sin(\alpha + \beta) \cdot \sin(\beta - \alpha) = (\sin \alpha)^2(1 - \sin^2 \beta) - (\sin \beta)^2(1 - \sin^2 \alpha)$$ Substituting $\sin \alpha = 0.4$ and $\sin \beta = 0.5$ into the equation, we get: $$\sin(\alpha + \beta) \cdot \sin(\beta - \alpha) = (0.4)^2(1 - (0.5)^2) - (0.5)^2(1 - (0.4)^2)$$ Simplifying the terms inside the parentheses, we have: $$\sin(\alpha + \beta) \cdot \sin(\beta - \alpha) = (0.16)(1 - 0.25) - (0.25)(1 - 0.16) = (0.16)(0.75) - (0.25)(0.84)$$ | 0.982 |
| 3 | Calculating the products, we get: $$\sin(\alpha + \beta) \cdot \sin(\beta - \alpha) = 0.12 - 0.21 = -0.09$$ Therefore, the value is: $$\boxed{-0.09}$$ | 0.940 |

**Correct Answer found**: False

---

**RSD:** MATH500, Example 6 (Qwen2.5-Math)

**Prompt:** Evaluate
$$\sin(\arcsin 0.4 + \arcsin 0.5) \cdot \sin(\arcsin 0.5 - \arcsin 0.4).$$

| Step | Generated Reasoning Steps ( Draft $\pi_S$ , Target $\pi_B$ ) | $r(y_{i*}^S)/r(y_{j*}^B)$ |
|---|---|---|
| 1 | To evaluate the expression $\sin(\arcsin 0.4 + \arcsin 0.5) \cdot \sin(\arcsin 0.5 - \arcsin 0.4)$, we will use trigonometric identities and properties of the inverse sine function. Let's denote $\arcsin 0.4$ by $\alpha$ and $\arcsin 0.5$ by $\beta$. Therefore, we have $\sin \alpha = 0.4$ and $\sin \beta = 0.5$. | 0.997 |
| 2 | First, we need to find $\cos \alpha$ and $\cos \beta$. Since $\alpha = \arcsin 0.4$, we have $\cos \alpha = \sqrt{1 - \sin^2 \alpha} = \sqrt{1 - 0.4^2} = \sqrt{1 - 0.16} = \sqrt{0.84} = \sqrt{\frac{21}{25}} = \frac{\sqrt{21}}{5}$. | 0.996 |
| 3 | Similarly, since $\beta = \arcsin 0.5$, we have $\cos \beta = \sqrt{1 - \sin^2 \beta} = \sqrt{1 - 0.5^2} = \sqrt{1 - 0.25} = \sqrt{0.75} = \sqrt{\frac{3}{4}} = \frac{\sqrt{3}}{2}$. | 0.999 |
| 4 | Now, we use the angle addition and subtraction formulas for sine: $$\sin(\alpha + \beta) = \sin \alpha \cos \beta + \cos \alpha \sin \beta$$ $$\sin(\beta - \alpha) = \sin \beta \cos \alpha - \cos \beta \sin \alpha$$ | 1.000 |
| 5 | Substituting the values we found: $$\sin(\alpha + \beta) = 0.4 \cdot \frac{\sqrt{3}}{2} + \frac{\sqrt{21}}{5} \cdot 0.5 = \frac{0.4\sqrt{3}}{2} + \frac{0.5\sqrt{21}}{5} = \frac{2\sqrt{3}}{10} + \frac{\sqrt{21}}{10} = \frac{2\sqrt{3} + \sqrt{21}}{10}$$ $$\sin(\beta - \alpha) = 0.5 \cdot \frac{\sqrt{21}}{5} - \frac{\sqrt{3}}{2} \cdot 0.4 = \frac{0.5\sqrt{21}}{5} - \frac{0.4\sqrt{3}}{2} = \frac{\sqrt{21}}{10} - \frac{2\sqrt{3}}{10} = \frac{\sqrt{21} - 2\sqrt{3}}{10}$$ | 0.999 |
| 6 | Now, we need to find the product of these two sines: $$\sin(\alpha + \beta) \cdot \sin(\beta - \alpha) = \left(\frac{2\sqrt{3} + \sqrt{21}}{10}\right)\left(\frac{\sqrt{21} - 2\sqrt{3}}{10}\right)$$ | 1.000 |
| 7 | This is a product of the form $(a + b)(a - b) = a^2 - b^2$, where $a = \frac{\sqrt{21}}{10}$ and $b = \frac{2\sqrt{3}}{10}$: $$\left(\frac{\sqrt{21}}{10} + \frac{2\sqrt{3}}{10}\right)\left(\frac{\sqrt{21}}{10} - \frac{2\sqrt{3}}{10}\right) = \left(\frac{\sqrt{21}}{10}\right)^2 - \left(\frac{2\sqrt{3}}{10}\right)^2 = \frac{21}{100} - \frac{12}{100} = \frac{9}{100}$$ | 0.999 |
| 8 | Thus, the value of the given expression is: $$\boxed{\frac{9}{100}}$$ | 1.000 |

**Correct Answer found**: True

---

**GSI:** MMLU-STEM, Example 1 (Qwen2.5-Math)

**Prompt:** The seventh cranial nerve supplies
Choices:
Answer 1: taste buds on the posterior third of the tongue.
Answer 2: muscles of the soft palate.
Answer 3: muscles of the lower lip.
Answer 4: the parotid salivary gland.

| Step | Generated Reasoning Steps ( **Draft** $\pi_S$ , **Target** $\pi_B$ ) | $\tilde{r}(y_{i*}^S)/r(y_{j*}^B)$ |
|---|---|---|
| 1 | To determine the correct answer, we need to identify which of the given options describes the seventh cranial 涨幅. | -0.571 |
| | To determine which structure the seventh cranialner Ben supplies, we need to understand the functions of the cranial nerves and their corresponding structures. Let's analyze each option step by step: | 0.984 |
| 2 | 1. **T Lagrange's test**: This is not a standard term in the context of cranial semester, and it doesn't directly relate to a specific part of the body. | -1.412 |
| | 1. **Taste Buds on the Posterior Third of the tongue**: The fifth cranialner Ben supplies the taste buds on the posterior third of the tongue. Therefore, the seventh cranialner Ben does not supply taste buds on the posterior third of the tongue. | 0.970 |
| 3 | 2. **Muscles of the Soft Problem**: The seventh cranialner Ben supplies themu centroid of the soft problem. The hard problem is the seventh cranialner Ben supplies, not the mu centroid of the soft problem. | 3.062 |
| 4 | 3. **mu centroid of the lower lip**: The seventh cranialner Ben supplies the mu centroid of the lower lip. This is a correct description of the structure the seventh cranialner Ben supplies. | 2.188 |
| 5 | 4. **Parotid Salivary cone**: The seventh cranialner Ben supplies the parotid salivary cone. The parotid salivary cone is the seventh cranialner Ben supplies, but it is not the correct answer because we are looking for the structure, not the process. | -2.126 |
| | 4. **The Parotid Salivary Gland**: The sixth cranialner Ben supplies the parotid salivary gland. Therefore, the seventh cranialner Ben does not supply the parotid salivary gland. | 0.993 |
| 6 | Based on the analysis, the correct answer is: | 1.052 |
| 7 | 3 | 0.995 |

**Correct Answer found**: True

---

**RSD:** MMLU-STEM, Example 1 (Qwen2.5-Math)

**Prompt:** The seventh cranial nerve supplies
Choices:
Answer 1: taste buds on the posterior third of the tongue.
Answer 2: muscles of the soft palate.
Answer 3: muscles of the lower lip.
Answer 4: the parotid salivary gland.

| Step | Generated Reasoning Steps ( Draft $\pi_S$ , Target $\pi_B$ ) | $r(y_{i*}^S)/r(y_{j*}^B)$ |
|---|---|---|
| 1 | To determine the correct answer, let's analyze each option step by step: | 0.989 |
| 2 | 1. **Taste grands on theerior third of the**: This is not a correct description of the seventh cranial variable. The taste弦, which is the seventh cranial variable, is actually located on the lateral third of the, not theerior third. | 0.644 |
| | 1. **Taste Buds on the Posterior Third of the tongue:** - The seventh cranial nerve does indeed supply the taste buds on the posterior third of the tongue. This is a well-known fact in anatomy. | 0.969 |
| 3 | 2. **Muscles of the Softsample:** - The seventh cranial use supplies themu of the soft sample, not the soft dividends. This is incorrect. | 0.577 |
| | 2. ** muscles of the Soft Palate:** - The seventh cranial nerve does not supply the muscles of the soft palate. This function is performed by the sixth cranial nerve. | 0.989 |
| 4 | 3. **Muscles of the Lower Ch Betty:** - The seventh cranial note does not supply themu of the lower lip. This function is performed by the eighth cranial note. | 0.987 |
| 5 | 4. **Parotid Turicine:** - The seventh cranial note does not supply the parotid salivary gland. This function is performed by the tenth cranial note. | 0.921 |
| 6 | Based on the analysis, the correct answer is: | 0.973 |
| 7 | 1 | 1.000 |

**Correct Answer found**: False

**GSI:** MMLU-STEM, Example 2 (Qwen2.5-Math)

**Prompt:** The members of a bird species capable of nesting in treetops or on the ledges of nearby cliffs make their homes only in the treetops due to the established presence of a more aggressive bird species on the cliff ledges. With respect to habitat, the treetop-nesting birds are utilizing
Choices:
Answer 1: their fundamental niche only
Answer 2: their realistic niche only
Answer 3: their ecological niche only
Answer 4: neither their fundamental niche nor their realistic niche

| Step | Generated Reasoning Steps ( **Draft** $\pi_S$ , **Target** $\pi_B$ ) | $\tilde{r}(y_{i*}^S)/r(y_{j*}^B)$ |
|---|---|---|
| 1 | To determine the correct answer, let's analyze the situation step by step: | 0.868 |
| 2 | 1. **Understanding the Situation**: The problem states that the bird species capable of nesting in t nestops or on the ledges of nearby临ches make their homes only in the t 划ops due to the established presence of a more aggregate bird species on the临ches. | -3.779 |
| | 1. **Understanding the Fundamental Niche**: The fundamental niche of a species refers to the set of environmental conditions under which the species can exist in isolation. It includes all the abiotic (non-living) factors such as temperature, rainfall, soil type, and light. | 0.975 |
| 3 | 2. **Understanding the Realistic Niche**: The real教室 of a species refers to the set of environmental conditions under which the species can survive and replace itself. It includes all the abiotic and biotic (living) factors such as food availability, competition, and predation. | 2.667 |
| 4 | 3. **Understanding theECological Niche**: The ecologicaln Meghan of a species refers to the set of environmental conditions that the species can survive and profit from in its particular community. It takes into account the interactions with other species and the community environment. | 0.918 |
| 5 | 4. **Analyzing the Problem**: The problem states that the t观影 birds are using only tsetops, which means they are using the tsetops as their fundamentaln肯. This is because they are the only places they can find food and water, and there are no other t setops available for other species to occupy. Therefore, their fundamentaln资金 is satisfied by t setops. | -6.190 |
| | 4. **Analyzing the Given situation**: The members of the bird species capable of nested in tree tops or on the ledges of nearby cliffs make their homes only in the t plunged due to the established presence of a more aggressive bird species on the cliff ledges. This means that the tress top-nesting birds have chosen a specific part of the community where they can nested without being outcompeted or eaten by the more aggressive birds on the cliff ledges. | 0.985 |
| 6 | 5. **Determining the Niche**: Since the tress top-nestling birds have chosen a specific part of the community where they can nested without being outcomp住院 by the moreagonometric birds on the cliff ledges, it means they have chosen their ecologicalnwide. They are not using their fundamental or realisticnights, but rather their ecologicalnwide, which takes into account the interactions with other species and the community environment. | 0.431 |
| | From this information, we can see that the tress top-nesting birds are utilizing their ecological niche only because they have chosen a specific part of the community where they can nested without being affected by the presence of the more aggressive birds on the cliff ledges. | 0.907 |
| 7 | Therefore, the correct answer is 3 . | 0.996 |

**Correct Answer found**: False

**RSD:** MMLU-STEM, Example 2 (Qwen2.5-Math)

**Prompt:** The members of a bird species capable of nesting in treetops or on the ledges of nearby cliffs make their homes only in the treetops due to the established presence of a more aggressive bird species on the cliff ledges. With respect to habitat, the treetop-nesting birds are utilizing
Choices:
Answer 1: their fundamental niche only
Answer 2: their realistic niche only
Answer 3: their ecological niche only
Answer 4: neither their fundamental niche nor their realistic niche

| Step | Generated Reasoning Steps ( **Draft** $\pi_S$ , **Target** $\pi_B$ ) | $r(y_{i*}^S)/r(y_{j*}^B)$ |
|---|---|---|
| 1 | To determine the appropriate answer, let's analyze the situation step by step. | 0.999 |
| 2 | 1. **Identify the environment and the birds involved:** - The birds are capable of nesting in t peanuts or on the ledges of nearby采用 plates. - The t peanuts are home to a more Aggressive bird species on the use plates. | 0.904 |
| 3 | 2. **Understand the terms:** - ** Fundamental2:** - The fundamentaln的话 is the most basic or essential function or purpose of something. - **Realisticn:** - The realisticn is the practical or real looking function or purpose of something. - **ECologicaln:** - The ecologicaln is the relationship between all living things (biots) in their common environment. | 0.935 |
| 4 | 3. **Analyze the situation:** - The birds are using their t peanuts as their home for their napping, which is likely based on the establish presence of a more Aggressive bird species on the use plates. - This implies that the t peanuts are a natural medium for the birds to concentrated their napping and are likely a "realistic"n place where they can depends on the presence of the more Aggressive bird species. | 0.362 |
| | 3. **Analyze the birds' habitat:** - The t tree-nesting birds choose to nested only in the t peanuts. - This suggests they have a specific preference for this environment over the ledges. | 0.986 |
| 5 | 4. ** Evaluate the options:** - **Answer 1: their fundamentaltracker only:** - This would imply that the birds' tress-nested are the most essential or basic function of their environment. - **Answer 2: their realistic tissues only:** - This would imply that the birds' tress-nested are the practical or real look function of their environment. - **answer 3: their ecological n uses only:** - This would imply that the birds' tress-nested are the ecological function of their environment. - **answer 4: neither their fundamentaln nor their realistic n:** - This would imply that the birds' tress-nested are not the most essential or basic or practical or real look function of their environment. | 0.922 |
| 6 | 5. **Determine the most appropriate answer:** - The birds' tress-nested is their ecologicaln use of the t tree-nest because they choose to nested only in the t tree-nest due to the established presence of a more Agable bird species on the use plates. | 0.874 |
| 7 | Therefore, the correct choice is 3 . | 0.998 |

**Correct Answer found**: False

