# OpenReview forum: "Guided Speculative Inference for Efficient Test-Time Alignment of LLMs"
_ICLR.cc/2026/Conference — ICLR 2026 Poster_

### Official Review · Reviewer_j8aM · 2025-10-31

**Soundness:** 2
**Presentation:** 3
**Contribution:** 3
**Rating:** 4
**Confidence:** 2

**Summary:**

The paper introduces Guided Speculative Inference (GSI), an efficient decoding algorithm that integrates reward-guided generation with speculative sampling from a smaller auxiliary model. GSI provably approximates the optimal soft best-of-n policy and achieves higher accuracy on reasoning benchmarks than both standard soft best-of-n and prior reward-guided speculative decoding methods.

**Strengths:**

1. The paper is well written and easy to follow.
2. The proposed approach is theoretically grounded, and the theoretic analysis holds under more realistic conditions compared to SPECS.
3. GSI empirically achieves better performance than other reward-guided speculation approaches, when evaluated on a wide range of math benchmarks.

**Weaknesses:**

The authors evaluate on only a single pair of LLMs and a single PRM. This makes the practical generalizability of their findings questionable. Also, the evaluation is heavily focused on math benchmarks.

**Questions:**

Could you provide some more results on diverse setup, including different LLMs, PRMs, and non-mathematic benchmarks?

---

> ### Author Response · Authors · 2025-11-22
> **Author Response to Reviewer j8aM**
>
> Thank you for your feedback! We address your main concern below.
>
> > **The authors evaluate on only a single pair of LLMs and a single PRM. This makes the practical generalizability of their findings questionable. Also, the evaluation is heavily focused on math benchmarks. Could you provide some more results on diverse setup, including different LLMs, PRMs, and non-mathematic benchmarks?**
>
> In the updated version of the paper, we have included an additional complete set of experiments using the Qwen3 model suite, with Qwen3-1.7B as the draft and Qwen3-14B as the target model. If time permits, we will also include additional experiments with other PRMs in the final version of the paper; however, we have not had the computational resources to add them to the updated version yet.
>
> We would also like to point out that the MMLU-STEM dataset is *not* a math dataset, and covers areas such as biology, physics, astronomy, chemistry, computer science, and electrical engineering. Please have a look at Appendix C.7 for a few sample prompts and responses.
>
> We hope this resolves all of your concerns about the paper!

---

### Official Review · Reviewer_KZqA · 2025-11-01

**Soundness:** 3
**Presentation:** 1
**Contribution:** 1
**Rating:** 2
**Confidence:** 4

**Summary:**

This paper proposes Guided Speculative Inference (GSI), a test-time algorithm that mixes (i) soft best of n (S-BoN), ii) a process reward model PRM $r$ and iii) speculative drafts from a small model $\pi_s$ to approximate the reward tilted distibution of a larger base/target model $\pi_B$. The method computes the tilted rewards as a linear combination of the rewards of generated seequences and loglikelihood ratios of those sequences between the target (big) model and the draft (small) model. The paper claims theoretical guarantees (KL bound to $\pi_{\beta,B}$ which is their approximation (GSI) algorithm, and empirical gains on math-reasoning benchmarks (MATH500, OlympiadBench, Minerva, MMLU-STEM, GSM8K) against RSD and S-BoN (with small model) and states that rarely GSI even outperforms S-BoN with big model.

**Strengths:**

* The algorithmic description with pseudocode in Algorithm 1 is clear.
* Reasoning-trace visualizations (Fig. 3) aid intuition about reject/accept decisions.

**Weaknesses:**

* The biggest problem is that the paper claims efficiency but the presented algorithm is not efficient at all on hardware utilization. Paper serves three models on separate H100s; no FLOPs/GPU-seconds/tokens-per-second or memory footprint comparisons vs S-BoN($\pi_B$) is provided. It seems that GSI uses more memory footprint compared to  S-BoN($\pi_B$)  (almost 30-50% more given that for reasoning traces the KV-Cache dominates model parameters hence small model and large model usage are comparable). Nonetheless, GSI fails to outperform S-BoN($\pi_B$) on most benchmarks, thus there is no or little incentive to use GSI.

* Aside from higher burden on the hardware, GSI could claim they are efficient because of being faster. However there is no clear demonstration of GSI scaling better with the time spent for answering problems against the accuracy in those problems. The x-axis in Figure 1, Figure 4 and Figure 5 have $n$, but this is expected and do not convey much information. Nonetheless, the question arises: why would it be desirable to run GSI when someone can run S-BoN($\pi_B$) with fewer resources and get better results?
* Theorem 1 bound is practically vacuous at $\beta=20$ with $r \in [0,1]$. No empirical check on $\chi^2(\pi_B||\pi_S)$ or the coverage constant $C_\infty$. The guarantee does not meaningfully inform realistic $n$.

Overall, I do not think this paper will be practically relevant or desirable to implement, and theory seems even further away from being useful.

**Questions:**

* Report end-to-end metrics: GPU-seconds, FLOPs, tokens/sec, peak VRAM, and number of served models per method.
* Include curves of accuracy vs end-to-end latency under a fixed hardware budget; compare to S-BoN($\pi_B$) and best-of-n baselines.
* Can you quantify PRM calibration and correlation with ground truth correctness?
* Please provide the time-breakdown of different components in the algorithm: how much time is spent on big model and how much time is spent on small model generation.
* Compare all the frameworks here against just doing speculative decoding with the same Big and Small models and doing S-BoN.
* Please provide empirical estimates or upper bounds for $\chi^2(\pi_B||\pi_S)$ (or $C_\infty$) on your tasks; discuss practicality at $\beta=20$.If the bound is loose, what guidance does it give practitioners?

---

> ### Author Response · Authors · 2025-11-22
> **Author Response to Reviewer KZqA (Part 1/2)**
>
> Thank you for your time and valuable feedback to our paper. We will respond to your concerns and questions below.
>
> > **The biggest problem is that the paper claims efficiency but the presented algorithm is not efficient at all on hardware utilization. Paper serves three models on separate H100s; no [...] tokens-per-second or memory footprint comparisons vs S-BoN(pi_B) is provided. [...] GSI fails to outperform S-BoN() on most benchmarks, thus there is no or little incentive to use GSI. [...] Include curves of accuracy vs end-to-end latency [...] Please provide the time-breakdown of different components in the algorithm**
>
> We would like to respectfully point out that reward-guided inference always necessarily serves a target and a reward model. Similarly, speculative decoding always necessarily serves a draft and a target model. Hence, any speculative sampling algorithm for reward-guided inference has to serve three models at once; this is not a shortcoming of our algorithm, but rather the setting in which we apply it. While the implementation can vary, an implementation with vLLM will host each model on its own GPU.
>
> We would also like to highlight that this implies that vanilla speculative decoding is inherently more memory consuming than naive inference from the target model; speculative decoding trades off memory for latency. Our algorithm does the exact same: we have to pay the additional memory footprint of hosting a draft model, for which we gain significant gains in latency. In Table 1, we show that GSI can yield up to 50% gains in latency (in terms of steps generated per second) compared to s-BoN with the target model, while oftentimes matching (or even outperforming) it in terms of downstream accuracy, retaining roughly 97% of its accuracy (cmp. Tables 2 and 3). Hence, we would like to clarify that the appeal of using GSI does not stem from it outperforming s-BoN with the target model in terms of accuracy, but rather from GSI yielding significant gains in terms of latency, while almost retaining the accuracy of the target model.
>
> We have included additional latency results for our new experiments with Qwen3 models in Table 1, which now also features an additional “steps / s” column (which is a good proxy for tokens / s).
> Additionally, we have added Figure 4, which features a time breakdown w.r.t. the three models for each method.
>
> We would also like to highlight two additional points: First, the PRM verification of draft steps and the computation of log-likelihoods of draft steps under the target model could be fully parallelized, which we have not implemented in our code for simplicity, but which would yield additional latency gains for GSI. Second, as can be seen in Figure 4, a major contributor in the end-to-end latency of GSI is the PRM verification. This suggests that GSI’s performance relative to the target model could be further improved when selecting a smaller PRM. While we have not had the time to include additional experiments with smaller PRMs for this rebuttal, we plan on adding them in the future.
>
>
> > **Aside from higher burden on the hardware, GSI could claim they are efficient because of being faster. However there is no clear demonstration of GSI scaling better with the time spent for answering problems against the accuracy in those problems. [...] why would it be desirable to run GSI when someone can run S-BoN($\pi_B$) with fewer resources and get better results?**
>
> We would like to refer you to our response to your question above. In particular, we highlight again that GSI can save significant amounts of latency (up to 50%, see Table 1) while retaining most of the target model’s performance.

---

> > ### Author Response · Authors · 2025-11-22
> > **Author Response to Reviewer KZqA (Part 2/2)**
> >
> > > **Theorem 1 bound is practically vacuous at beta=20 with r in [0,1]. No empirical check on chi^2(pi_B||pi_S)  or the coverage constant C_inf. The guarantee does not meaningfully inform realistic n.
> > Please provide empirical estimates or upper bounds for chi^2(pi_B||pi_S) (or C_inf) on your tasks; discuss practicality at beta=20.If the bound is loose, what guidance does it give practitioners?**
> >
> > We agree that the upper bound in Theorem 1 can get large for larger $\beta$. However, note that Theorem 1 holds for *any* $\beta>0$; in particular, by choosing $\beta$ small enough, $\exp(2\beta ||r||)$ can be made arbitrarily close to $1$. In practice, this guarantees practitioners that they can effectively bound the KL to the target distribution by this quantity, and one could choose $\beta$ small to get a lower theoretical guarantee in practice; however, we recommend to choose $\beta=20$ as this yields even better downstream performance than smaller values of $\beta$, compare Appendix C.3.
> >
> > Regarding estimates for $\chi^2$ and $C_\infty$, we first note that the bound in Theorem 1 is independent of the constant $C_\infty$; Assumption 1 is necessary simply to ensure objects in the proof of Theorem 1 are well-defined. The important quantity for the bound in Theorem 1 is $\chi^2$. We have added an empirical estimation of $\chi^2$ to the updated version of the paper, please see Appendix C.5. In particular, we show that the $\chi^2$ is very well-behaved in practice with average values of between $1.5$ and $3.9$ depending on the model family, and that Theorem 1 could indeed be used to inform realistic bounds (although as mentioned above, we recommend using $\beta=20$ for practical reasons).
> >
> > > **Can you quantify PRM calibration and correlation with ground truth correctness?**
> >
> > As this question is more concerned with the PRM itself than our algorithm, we refer you to the original paper introducing the Qwen2.5-Math-PRM-7B model [1]. The paper shows that it has an F1 score between 0.7 and 0.8, depending on the dataset (cmp. Table 7 in that paper).
> >
> > > **Compare all the frameworks here against just doing speculative decoding with the same Big and Small models and doing S-BoN.**
> >
> > Thank you for pointing this out, as we realized this is something we have not adequately addressed in our paper. Vanilla speculative sampling is known to scale very poorly with batch size; even modern frameworks like EAGLE-2 [2] have been shown to have a token throughput that is 50% less than that of the target model at a batch size of $n=64$ [3]. In particular, even sophisticated frameworks like EAGLE-3, that require targeted finetuning of the draft model, have not been evaluated beyond $n=64$ and do not achieve strictly better throughput than the target model alone [4]. Thus, test-time scaling as in GSI is not compatible with vanilla speculative decoding. Note that GSI circumvents this issue altogether, as generation both from the draft, as well as the target model remains fully parallelizable. We have included a discussion of this in Section 5 of the updated version of the paper (under “Methods”).
> >
> > [1] Zhang, Z., Zheng, C., Wu, Y., Zhang, B., Lin, R., Yu, B., Liu, D., Zhou, J., & Lin, J. (2025). The Lessons of Developing Process Reward Models in Mathematical Reasoning. URL: https://arxiv.org/abs/2501.07301
> >
> > [2] Li, Y., Wei, F., Zhang, C., & Zhang, H. (2024). EAGLE-2: Faster Inference of Language Models with Dynamic Draft Trees. URL: https://arxiv.org/abs/2406.16858
> >
> > [3] Yan, S., Zhu, M., Jiang, G., Wang, J., Chen, J., Zhang, W., Liao, X., Cui, X., Zhang, C., Song, Z., & Zhu, R. (2025). Scaling Laws for Speculative Decoding. URL: https://arxiv.org/abs/2505.07858
> >
> > [4] Li, Y., Wei, F., Zhang, C., & Zhang, H. (2025). EAGLE-3: Scaling up Inference Acceleration of Large Language Models via Training-Time Test. URL: https://arxiv.org/abs/2503.01840

---

### Official Review · Reviewer_8re2 · 2025-11-01

**Soundness:** 3
**Presentation:** 3
**Contribution:** 3
**Rating:** 6
**Confidence:** 3

**Summary:**

- The paper presents an interesting approach for efficient reasoning by combining speculative decoding with reward guidance.
- Demonstrates improved accuracy over state-of-the-art baselines, though at the cost of higher compute.
- The paper is well-written, and the proofs are well-justified.

**Strengths:**

- Novel integration of speculation and reward guidance for reasoning tasks.
- Strong empirical results showing accuracy improvements.
- Theoretical analysis and proofs are clear and rigorous.
- Writing quality is high, making the paper easy to follow.

**Weaknesses:**

- The scaling factor $\beta$ (to balance reward and log-likelihood ratio) requires exhaustive search, which reduces practical efficiency.
- Higher computational cost compared to standard RSD methods.
- Limited discussion on scalability and efficiency trade-offs.

**Questions:**

1. The draft model appears to perform strongly based on Table 1, particularly as $n$ increases. Could the authors provide an analysis of the proposed method’s effectiveness when using weaker draft models?
2. For draft models with higher $n$, performance seems competitive. Could the authors report the accuracy for the s-BON draft configuration with $n=16$?
3. How is the activation point for the PRM model determined? or how many tokens are generated before calling the PRM. What heuristic is this based on? For reference in speculative decoding, the optimal draft length is downstream task dependent.
4. The new reward formulation incorporates the logarithm of the acceptance criterion. Could the authors elaborate on any theoretical connections between this formulation and standard speculative decoding?
5. The experiments utilize Qwen2.5, while newer and stronger models such as Qwen3 are available. Could the authors clarify why Qwen3 was not considered?
6. The PRM model is reported to be the same size as the base model. Were any ablation studies conducted with smaller PRM models to assess trade-offs between performance and computational cost?
7. In Theorem 2, expression (b), is there a missing term involving $\tilde{\pi}_{\text{GSI}}(Y\geq)$​ ?
8. In Figure 6, each curve appears to correspond to a different value of β\betaβ. Do the circular points on the curves represent specific β\betaβ values? If so, please clarify.
9. The paper mentions that Table 2 compares GSI with SPECS, yet there is no row for SPECS in the table. Could the authors explain this discrepancy?

---

> ### Author Response · Authors · 2025-11-22
> **Author Response to Reviewer 8re2 (Part 1/2)**
>
> Thank you for your positive and helpful feedback!
> We answer your questions and concerns below.
>
> > **The scaling factor beta  (to balance reward and log-likelihood ratio) requires exhaustive search, which reduces practical efficiency.**
>
> While it is a priori unclear what would be the best value for $\beta$, we added additional experiments with the Qwen3 model family that show that $\beta=20$ works very well out-of-the-box on these models as well. Furthermore, we would like to highlight that we only calibrated $\beta$ on MATH500 (compare Figure 7), and $\beta=20$ works on all other datasets without further search. These results show that $\beta=20$ can be used out-of-the-box without further search both across model families, as well as across datasets.
>
> EDIT: We have now added an additional ablation over $\beta$ for our new Qwen3 experiments, which shows that $\beta=20$ is indeed optimal for Qwen3; see Figure 8 of the revised PDF.
>
> > **[GSI suffers from] [h]igher computational cost compared to standard RSD methods.**
>
> RSD is indeed more computationally efficient, as it does not require computing log-likelihoods of draft steps under the target model. However, as we show in our paper (cmp. e.g. Figure 2), RSD does not actually yield any meaningful improvements over the draft model (while being computationally more expensive), so the more relevant baseline to compare against is soft best-of-$n$ sampling with the target model $\pi_B$. As we show e.g. in Table 1 of the revised version of the paper, GSI increases throughput compared to soft best-of-$n$ with $\pi_B$ by up to $50\%$ while retaining $97\%$ of its accuracy, while RSD drops to $88\%$ of its accuracy.
>
> > **Limited discussion on scalability and efficiency trade-offs.**
>
> Could you please clarify what exactly you mean with scalability and efficiency trade-offs? We scale all our experiments up to $n=256$, which is quite high; in particular, you can see e.g. in Figure 2 or Tables 2 and 3 that the performance of both the draft and target model almost plateaus in that regime. Furthermore, we discuss efficiency trade-offs in Section 5.1, including new latency evaluations of our newly added Qwen3 experiments (Table 1 and Appendix C.6) and a new runtime breakdown in Figure 4. If there is a particular aspect that you believe we should discuss in more detail, please let us know!
>
> > **The draft model appears to perform strongly based on Table [2]*, particularly as $n$ increases. Could the authors provide an analysis of the proposed method’s effectiveness when using weaker draft models?**
>
> You are right that the performance gap between Qwen2.5-Math-1.5B-Instruct and Qwen2.5-Math-7B-Instruct is, on average, not very large. We included additional experiments with Qwen3-1.7B as the draft, and Qwen3-14B as the target model. As you can see in Table 1, Figure 2, and Table 3, these models exhibit a more significant gap between draft and target, and GSI performs even stronger on this model family.
>
> *Table number in your question adjusted to reflect the updated version of the paper.
>
> > **For draft models with higher n, performance seems competitive. Could the authors report the accuracy for the s-BON draft configuration with n=16?**
>
> We already report the performance of s-BoN with the draft model in all our experiments, including for $n=16$ (compare e.g. Figure 2 or Table 1). If you were referring to something else, please clarify what you meant!
>
> > **How is the activation point for the PRM model determined? or how many tokens are generated before calling the PRM. What heuristic is this based on? For reference in speculative decoding, the optimal draft length is downstream task dependent.**
>
> We always generate complete *reasoning steps* from the draft model, where a reasoning step ends when the model generates a double line break “\n\n”. These steps typically contain between 10-100 tokens. However, this step is fully automated: There’s no manual step length that we set a priori for our algorithm. While we briefly explained this towards the end of Section 4 in the paper, we realized we did not make it clear enough, so we included a more detailed explanation in the revised paper in Section 4, and also in Algorithm 1.

---

> > ### Author Response · Authors · 2025-11-22
> > **Author Response to Reviewer 8re2 (Part 2/2)**
> >
> > > **The new reward formulation incorporates the logarithm of the acceptance criterion. Could the authors elaborate on any theoretical connections between this formulation and standard speculative decoding?**
> >
> > Our algorithm and standard speculative decoding are conceptually very different; note, in particular, that vanilla speculative decoding does not have any reward model or rewards, as it is merely concerned with sampling from the target model’s distribution, not doing so in a reward-maximizing way. This allows for a simple rejection criterion in speculative decoding that guarantees that one provably samples exactly from the target model’s distribution, where the rejection criterion is defined in terms of the ratio $\pi_B(y|x)/\pi_S(y|x)$ [1]. However, in our setting, it is not possible to obtain a comparable theoretical guarantee, as the target distribution we try to approximate – the tilted distribution $\pi_B(y | x) \exp(\beta r(x,y)) / Z_{\beta,B}(x)$, see Equation (1) in the paper – cannot be sampled from directly. Incorporating the log-ratio of the likelihoods under both models, as we do in our tilted rewards, ensures that we at least stay close to the tilted target distribution *asymptotically*, as shown in Theorem 1. This can be seen as replacing the likelihood ratio $\pi_B(y|x)/\pi_S(y|x)$ in vanilla speculative decoding with the tilted ratio $\frac{\exp(\beta r(x,y))}{Z_{\beta,B}(x)}\pi_B(y|x)/\pi_S(y|x)$.
> >
> > > **The experiments utilize Qwen2.5, while newer and stronger models such as Qwen3 are available. Could the authors clarify why Qwen3 was not considered?**
> >
> > We have included additional experiments with the Qwen3 model family in the revised version of the paper, specifically with Qwen3-1.7B as the draft, and Qwen3-14B as the target model; please see Table 1, Figure 2, or Table 3 for the new results.
> >
> > > **The PRM model is reported to be the same size as the base model. Were any ablation studies conducted with smaller PRM models to assess trade-offs between performance and computational cost?**
> >
> > Initially, we experimented with various PRMs and found Qwen2.5-Math-PRM-7B to be both strong and compatible with vLLM, which is why we chose it for our experiments.
> > While we do not have the computational resources to include another ablation over the size of the PRM for this rebuttal, we refer you to our new set of experiments on the Qwen3 model family, where we use the same PRM as before. Since our target model in these experiments is of size 14B, this shows that GSI works equally well, if not better, when the PRM is smaller than the target model. If time permits, we will include an additional set of experiments with another PRM of a different size in the final version of the paper.
> >
> > > **In Theorem 2, expression (b), is there a missing term involving $\tilde{\pi}(Y_{\ge})$?**
> >
> > Thank you for checking our proofs and for bringing this to our attention! The derivation is correct; however, we noticed that the way we wrote it down makes it a bit difficult to follow. We rewrote that derivation in the proof of Theorem 2 and also added color-coding to the different terms to make it more readable; please let us know if this clarifies things!
> >
> > > **In Figure 6, each curve appears to correspond to a different value of [β]. Do the circular points on the curves represent specific [β] values? If so, please clarify.**
> >
> > The circular points on each curve correspond to different values $n$, specifically to $n=1,4,16,64,256$ (i.e. the first point on each curve corresponds to $n=1$, the second one to $n=4$, etc.). We have updated the description of the figure to clarify this.
> >
> > > **The paper mentions that Table [1] compares GSI with SPECS, yet there is no row for SPECS in the table. Could the authors explain this discrepancy?**
> >
> > We carefully checked every reference to Table 1, but we don’t mention anywhere that this table compares to $\texttt{SPECS}$. In particular, as $\texttt{SPECS}$ does not have a publicly available implementation, we were unable to compare it with GSI in terms of runtime.
> >
> > *Table number in your question adjusted to reflect the updated version of the paper.
> >
> > [1] Leviathan, Y., Kalman, M., & Matias, Y. (2023). Fast Inference from Transformers via Speculative Decoding. In Proceedings of the 40th International Conference on Machine Learning (ICML 2023), Honolulu, Hawaii, USA. PMLR 202. URL: https://arxiv.org/abs/2211.17192

---

### Author Response · Authors · 2025-11-22
**General Response by the Authors**

We thank all the reviewers for their time and helpful feedback. We appreciate how reviewers think our paper has “strong empirical results” (**8re2**), is “well written” and “easy to follow” (**j8aM**, **8re2**), and “theoretically grounded” (**j8aM**) with “[rigorous] theoretical analysis” (**8re2**).

Based on the reviewer’s feedback, we have **uploaded a revised PDF of the paper** with additional experiments and various additional clarifications. All **changes in the revised PDF are marked in blue**. In the following, we **address some common themes of the reviews** and **summarize additional experiments** contained in the updated version of the paper:

- **Experiments with additional model family:** We have added a new complete set of experiments with the Qwen3 model family, using Qwen3-1.7B as the draft, and Qwen3-14B as the target model. Our results show that GSI leads to even more significant speedups with Qwen3, e.g. **for $n=16$ we observe a 50% increase in generation throughput while retaining 97% of the performance of the target model (see Section 5.1).** We will try to include additional experiments with another model family and a smaller PRM in the camera ready version of the paper.

- **Computational complexity and latency:** We provide additional results regarding the latency of GSI, such as Table 1 which now includes a latency breakdown of the Qwen3 models, and Figure 4, which breaks down the latency of each decoding method into the three models. Furthermore, we added Section C.5 which discusses Theorem 1 and its practical implications in more detail.

In addition, we made several clarifications throughout the paper which address all remaining concerns of the reviewers, including a more detailed description of our draft step in Section 4 (in response to reviewer **8re2**); noting that $\beta$ does not require additional search in Section 5.3 (in response to reviewer **8re2**); a discussion of vanilla speculative decoding, and why it cannot be used in our setting in Section 5 (in response to reviewer **KZqA**); and a more detailed proof of Theorem 2 (in response to reviewer **8re2**).

---

> ### Author Response · Authors · 2025-11-27
> **Updated PDF with additional experiments**
>
> We have uploaded a revised version of the PDF, which now contains any remaining experiments. This includes complete results of the Qwen3 models on OlympiadBench (which were previously not yet available), as well as an **ablation over $\beta$ for Qwen3, which shows that our choice of $\beta=20$ is clearly optimal (even more distinctly than with Qwen2.5-Math models); cmp. Figure 8.** This addresses the concern by reviewer **8re2** about how to find the optimal $\beta$, and shows that no additional search is needed.

---

### Meta-Review · Area_Chair_bX8n · 2026-01-15

**Summary:**

This paper introduces Guided Speculative Inference (GSI), a novel algorithm for test-time scaling that combines speculative decoding with soft best-of-n sampling. The submission is notable for offering the first provable distributional guarantees for reward-guided speculation, approximating the optimal policy as sample size increases. Empirically, GSI achieves significant efficiency gains, reducing end-to-end latency by up to 28% and increasing throughput by 50% in updated Qwen3 experiments, all while maintaining high accuracy.

**Reviewer Concerns:**

Although one reviewer recommended rejection due to hardware efficiency concerns , the authors satisfactorily addressed this by demonstrating that the increased memory footprint is an accepted trade-off for the significant latency reductions achieved. The combination of rigorous theoretical analysis and robust empirical results justifies acceptance

**Reviewer Scores:**

Although one reviewer recommended rejection due to hardware efficiency concerns , the authors satisfactorily addressed this by demonstrating that the increased memory footprint is an accepted trade-off for the significant latency reductions achieved. The combination of rigorous theoretical analysis and robust empirical results justifies acceptance

---

### Decision · Program_Chairs · 2026-01-26

Accept (Poster)